# Reconstruction of track and simulation of storm surge associated with the calamitous typhoon affecting the Pearl River Estuary in September 1874

Hing Yim MOK, Wing Hong LUI, Dick Shum LAU, Wang Chun WOO

Hong Kong Observatory

## Abstract

A typhoon struck the Pearl River Estuary in September 1874 (the "Typhoon 1874"), causing extensive damages and claiming thousands of lives in the region during its passage. Like many other historical typhoons, the deadliest impact of the typhoon was its associated storm surge. In this paper, a possible track of the typhoon was reconstructed by analysis of the historical qualitative and quantitative weather observations in the Philippines, the northern part of the South China Sea, Hong Kong, Macao and Guangdong recorded in various historical documents. The magnitudes of the associated storm surges and storm tides in Hong Kong and Macao were also quantitatively estimated using storm surge model and analogue astronomical tides based on the reconstructed track. The results indicated that the typhoon could have crossed the Luzon Strait from the western North Pacific and moved across the northeastern part of the South China Sea to strike the Pearl River Estuary more or less as a super typhoon in the early morning on 23 September 1874. The typhoon passed about 60 km south-southwest of Hong Kong and made landfall in Macao, bringing maximum storm tides of around 4.9 m above the Hong Kong Chart Datum[1] at the Victoria Harbour in Hong Kong and around 5.4 m above the Macao Chart Datum[2] at Porto Interior (inner harbour) in Macao. Both the maximum storm tide (4.88 m above Hong Kong Chart Datum) and maximum storm surge (2.83 m) brought by Typhoon 1874 at the Victoria Harbour estimated in this study are higher than all the existing records since the establishment of the Hong Kong Observatory in 1883, including the recent records set by super typhoon Mangkhut on 16 September 2018.

Keywords: 1874, typhoon, storm surge, storm tide, Hong Kong, Macao

---

[1] - http://www.geodetic.gov.hk/smo/gsi/Data/pdf/explanatorynotes.pdf

[2] - https://mosref.dscc.gov.mo/Help/ref/Macaucoord_2009_web_EN_v201702.pdf

## 1. Introduction

Hong Kong, located on the coast of southern China, is vulnerable to sea flooding due to storm surges associated with approaching tropical cyclones from the western North Pacific or the South China Sea.  Since the establishment of the Hong Kong Observatory in 1883 when records of tropical cyclones affected Hong Kong began, storm surges induced by typhoons in 1906, 1937 and Super Typhoon[3] Wanda in 1962 brought severe casualties and damages to Hong Kong (Peterson, 1975; Ho, 2003).  Storm surges induced by Super Typhoon Hope in 1979 (HKO, 1980), Severe Typhoon Hagupit in 2008 (HKO, 2009), Super Typhoon Hato in 2017 (https://www.hko.gov.hk/informtc/hato17/hato.htm; Lau & Chan, 2017) and Super Typhoon Mangkhut in 2018 (https://www.hko.gov.hk/blog/en/archives/00000216.htm), even though with no significant casualties, still brought severe flooding and damages to Hong Kong during their passages.  Storm surge and sea level records, as well as the tracks of these typhoons, are shown in Table 1 and Figure 1 respectively.

Figure 1 shows that typhoons bringing significant storm surge impact to Hong Kong have similar tracks - forming and intensifying into at least typhoon strength over the western North Pacific, moving across the Luzon Strait without making landfall over the Philippines and Taiwan (except Mangkhut which skirted the northern part of Luzon), approaching the coast of Guangdong and making landfall over or passing to the south of Hong Kong.  Typhoons moving across the Luzon Strait without making landfall over the Philippines and Taiwan can maintain their intensity, and making landfall over or passing to the south of Hong Kong will generate onshore winds that bring severe storm surges to the territory.

However, for a better understanding of the storm surge risk in Hong Kong from a historical perspective, one should not ignore the calamitous typhoon which struck the Pearl River Estuary during 22 - 23 September 1874 (hereafter "Typhoon 1874"), bringing extensive damages and claimed several thousand lives in Hong Kong, and might have prompted the establishment of the Hong Kong Observatory

[3] - Classification of tropical cyclones in Hong Kong in terms of maximum sustained wind speeds near the centre averaged over a period of 10 minutes can be found at https://www.weather.gov.hk/informtc/class.htm

(the official weather authority in Hong Kong) later in 1883. "Hong Kong Typhoons" (Heywood, 1950) recorded that "the typhoon demolished the Civil Hospital and St. Joseph's Church (the locations are marked in Figure 2). A warship dragged her moorings and was thrown into V.R.C. boathouse (the location is marked in Figure 2)". As described in the report by the Captain Superintendent of Police (Hong Kong Government, 1874), "The Police had recovered the bodies of 621 people, but this number probably represented only one third of the actual figure. Furthermore, over 200 houses were destroyed or rendered uninhabitable. Two steamers sank in the harbour and another steamer was on shore near Aberdeen (the location is marked in Figure 2), and eight ships were supposed to have been lost. It was impossible to estimate the destruction of junks and small boats. Telegraph posts were blown down in different parts of the Hong Kong Island, interrupting communications. The roads were almost impassable from the obstruction caused by the fallen trees." The China Mail of 23 September 1874 (China Mail, 1874) also reported that "A typhoon, though of very short duration, has probably proved the most destructive witnessed since 1862 – if not exceeding it in that respect - swept over the island between the hours of 6 p.m. and 6 a.m.".

Like many other historical typhoons which caused significant casualties, the deadliest impact of Typhoon 1874 was its associated storm surge. The Harbour Master reported in the Hong Kong Government Gazette of 17 October 1874 (Hong Kong Government, 1874) that "The strength of the wind brought an immense volume of water into the harbour, not a tidal wave, but a rapid rise which continued for about an hour, flooding the Praya (the waterfront of the northern part of the Hong Kong Island coloured in green in Figure 2) and ground floors of houses to a height of 4 and 5 feet for some distance inshore. Although, according to ordinary calculation it should have been low water at two o'clock; by three, the water had risen to from five to six feet above its high water level, or a rise of about ten feet had taken place."

Typhoon 1874 also caused severe damage to Macao on the western side of the Pearl River Estuary. According to the publication "O Maior Tufao De Macau - 22 e 23 de Setembro de 1874" by Father Manuel Teixeira (1974), the passage of Typhoon 1874 was also accompanied by storm surge which caused severe flooding of up to 2.1 metres (7 feet) above the high tide level. "The Times of Typhoon" published by the Arquivo Historico De Macao (2014) summarized that "The strong winds, fierce waves and fires destructed countless buildings and

infrastructures, around 2,000 fishing vessels and cargo ships were sunk. The storm claimed around 5,000 lives, including approximately 3,600 in the Macao Peninsula, 1,000 in Taipa and 400 in Coloane (the locations are shown in Figure 3). The cost of the damage was estimated to be up to 2 million silver coins." Out of an estimated population of around 60,000 people in Macao (Ou Bichi (區碧池), 2014), the fatality accounted for approximately 8% of the population.

While Typhoon 1874 was one of the most damaging typhoons with significant storm surge in Hong Kong, no official records of any kind were available. Reconstruction of the track of the typhoon and conducting a quantitative estimation of its associated storm surge and storm tide during its passage based on the reconstructed track is therefore highly desirable for comparison with the other historical typhoons which affected Hong Kong, in particular, those which had generated significant storm surges listed in Table 1, and more importantly for assessing the storm surge risk in Hong Kong.

The objective of this study is to reconstruct the lifespan of Typhoon 1874 and to estimate quantitatively the storm surges and storm tides which might have been experienced in Hong Kong during its passage. The storm surges and storm tides in Macao are also estimated quantitatively for comparison in this study.

## 2. Data and Methods

A search and analysis of publicly available historical documents related to Typhoon 1874 was conducted to acquire the relevant information for reconstructing the track of the typhoon using the Jelesnianski tropical cyclone model (Jelesnianski, 1965). With the reconstructed track, the storm surges at Hong Kong and Macao during its passage were estimated by the storm surge model SLOSH (Sea, Lake, and Overland Surges from Hurricanes) developed by the National Oceanic and Atmospheric Administration (NOAA) of USA. The astronomical tides were estimated by analogue with dates of similar Sun-Earth-Moon configuration for estimation of the storm tides (sum of storm surges and astronomical tides).

## 3. Historical documents available for the study

After a search and study of historical documents, the following information available from the Philippines, Hong Kong and Macao provided the valuable weather and tidal observations for reconstructing the lifespan and revealing the power of Typhoon 1874:

130

 

(a) The Selga Chronology Part I available at http://www.ucm.es/info/tropical/selga-i.html (R. Garcia-Herrera, et al.) containing the wind and pressure records in the vicinity of the Luzon Strait during the passage of Typhoon 1874;

(b) The Harbour Master Report in the Hong Kong Government Gazette on 17 October 1874 (Hong Kong Government, 1874) containing the qualitative observations of three ships (namely British ship Onward, American ship Highlander, German ship Amanda) on Typhoon 1874 near the then Pratas Shoal (now called Dongsha as marked in Figure 1) over the northeastern part of the South China Sea, description of the sequence of weather changes in the Victoria Harbour and pressure readings at the Central Police Station (the location is marked in Figure 2) in Hong Kong during the passage of Typhoon 1874;

140

(c) The China Mail of 23 September 1874 (China Mail, 1874) and Hong Kong Daily Press of 24 September 1874 (Hong Kong Daily Press, 1874) containing the pressure readings reduced to mean sea level in the Victoria Harbour from the Harbour Master and pressure readings recorded by two barometers of a local company, Messers Falconer & Co., in Hong Kong (the location is marked in Figure 2) during the passage of Typhoon 1874;

150

(d) The Hong Kong Daily Press of 25 September 1874 (Hong Kong Daily Press, 1874) describing the impact of Typhoon 1874 to the then Swatow (now called Shantou as marked in Figure 1) at the eastern coast of the Guangdong Province of China;

(e) The logbook of the vessel HMS Princess Charlotte moored at the Victoria Harbour being kept at the National Archives in London, UK containing observations on pressure and winds for 22 and 23 September 1874;

160

(f) Bulletin of the Province of Macao and Timor published on 3 October 1874 (Macao Government, 1874) containing the ship reports on 22 to 23 September 1874 by gunboat Tejo moored at the Porto Interior (the location is shown in Figure 3) in Macao;

(g) Bulletin of the Province of Macao and Timor published on 24 October 1874 (Macao Government, 1874) containing the weather observations from 11 a.m. on 22 September to 10 a.m. on 23 September 1874 by the then Macao Port Authority; and

(h) A book on historical disastrous tidal events in China (Lu Renji (陸人驥), 1984) describing the track of Typhoon 1874 in the Pearl River Estuary and inland western Guangdong and its damages to the region.

## 4. Analysis of weather observations in historical documents

Construction of a possible track of Typhoon 1874 was divided into three parts in this study, namely, (a) over the Luzon Strait, (b) over the South China Sea, and (c) over the Pearl River Estuary.

(a)     Luzon Strait

Description of the passage of Typhoon 1874 in the vicinity of Luzon Strait was found in "The Selga Chronology Part I: 1348-1900" (R. Garcia-Herrera, *et al.*). Annex I is the extract of the part on Typhoon 1874 with description on the meteorological conditions at two observation points, Vigan (a city near the western coast of Luzon) and Batan Islands in the Luzon Strait during the passage of the typhoon.

Figure 4 shows the geographical location of Vigan and Batan Islands and Table 2 and Table 3 show the chronological summary of the meteorological observations at the two locations respectively based on the information in Annex I.   It can be seen that Batan Islands and Vigan were affected by hurricane force winds from the east and west respectively at around midnight of 21 September, suggesting that the centre of Typhoon 1874 was passing through the area between Vigan and Batan Islands at about that time.  As the pressure recorded at Batan Islands at

1 a.m. on 22 September (724 mmHg or 965.3 hPa) was much lower than that recorded at Vigan at midnight of 21 September (745 mmHg or 993.3 hPa), the centre of Typhoon 1874 was likely closer to Batan Islands than Vigan and located over the sea area off the coast of northern Luzon during its passage over the Luzon Strait, as indicated by the arrow in Figure 4.

Furthermore, the pressure of 724 mmHg (965.3 hPa) observed at Batan Islands at 1 a.m. on 22 September suggested that Typhoon 1874 had an intensity of at least a typhoon with maximum sustained 10-minute mean wind speed reaching 130 km/h or higher near its centre using the minimum pressure and maximum wind relationship for tropical cyclones in this region (Atkinson, 1977). Assuming the centre of Typhoon 1874 was closer to the Batan Islands, the observed hurricane winds at Vigan suggested that the radius of hurricane force winds of Typhoon 1874 was over 200 km when it passed through the Luzon Strait.

(b)     South China Sea

The description of Typhoon 1874 over the northeastern part of the South China Sea was found in the Harbour Master report in the Hong Kong Government Gazette published on 17 October 1874 (Hong Kong Government, 1874) which stated that the typhoon passed rather close to the then Pratas Shoal (now called Dongsha as marked in Figure 1) according to the reports from three ships (British ship Onward, American ship Highlander, and German barque Amanda) travelling close to the island between 4 and 6 p.m. on 22 September. However, detailed meteorological observations from these three ships could not be found for more in-depth interpretation such as whether the typhoon passed to the south or north of Dongsha as it approached the South China coast.

On the other hand, it was reported in the Hong Kong Daily Press of 25 September 1874 (Hong Kong Daily Press, 1874) that "there has also been a most severe typhoon at Swatow (now called Shantou) and the sea ran so high as to flood the Custom House, which is three hundred yards inland, to such an extent as to damage the whole of the papers in the office", indicating that Typhoon 1874 had also brought severe storm surge to Shantou before reaching the Pearl River Delta. Considering the distance of about 300 km between Dongsha and Shantou, Typhoon 1874 likely passed to the north rather than to the south of Dongsha.

(c)     Pearl River Estuary

A summary of the pressure observations in Hong Kong and Macao during the passage of Typhoon 1874 extracted from the available historical documents is shown in Table 4.  Figure 5 plots the time series of those pressure observations from 8 a.m. on 22 September to 11 a.m. on 23 September 1874.  It can be seen that the pressure at Hong Kong reached the minimum of around 975 hPa at 2 a.m. on 23 September while the pressure at Macao continued to fall to a minimum of around 946 hPa at 4 a.m. on 23 September.  Furthermore, the pressure readings and pressure falling rate at Hong Kong and Macao were rather close with each other from the evening of 22 September to 2 a.m. on 23 September.  This phenomenon revealed that Typhoon 1874 was at more or less the same distance from Hong Kong and Macao when it approached the Pearl River Estuary during this period.

A summary of the wind observations in Hong Kong and Macao during the passage of Typhoon 1874 extracted from the available historical documents is shown in Table 5.  In Hong Kong, the north to northwesterly winds started to strengthen in the evening on 22 September.  Winds continued to strengthen and veered gradually to east-northeast and reached hurricane force by 2 a.m. on 23 September.  Hurricane force winds maintained for the next two hours while the winds gradually veered to east-southeast.  The winds gradually subsided and veered to the southeast or south-southeast in the early morning on 23 September.

According to the result of a study on the relation between tropical cyclone position and wind direction at Waglan Island (an offshore island over the southeastern part of Hong Kong) during strong winds or above situations using tropical cyclones from 1968 to 2001 (Figure 6, Hong Kong Observatory), the sequence of wind direction change at Hong Kong during the passage of Typhoon 1874 suggested that Typhoon 1874 might most possibly approach Hong Kong from more or less the east and pass to the south of Hong Kong in the small hours on 23 September.  This also provided further support for Typhoon 1874 passing to the north of Dongsha during its passage in the northeastern part of the South China Sea.

In Macao, the winds changed in more or less the same way as those in Hong Kong in the evening on 22 September.  Hurricane force winds affected Macao from 3 a.m. to 5 a.m. on 23 September and there was a rapid change of wind direction of 90 degrees from northeast to southeast at 4 a.m.  The pressure and wind

observations in Table 4 and 5 respectively revealed that Typhoon 1874 possibly made landfall at Macao or pass rather close to the south of the gunboat Tejo moored at Porto Interior at 4 a.m. on 23 September with a minimum mean sea level pressure of around 945 hPa near its centre.  This minimum mean sea level pressure suggested that the maximum sustained 10-minute mean winds near the centre of Typhoon 1874 could have exceeded 180 km/h according to Atkinson (1977), indicating that the intensity of Typhoon 1874 might have reached the strength of a super typhoon (i.e. maximum sustained 10-minute mean wind speed of 185 km/h or more) when it made landfall at Macao.  Considering that the centre of Typhoon 1874 was at Porto Interior in Macao at 4 a.m. on 23 September, the marginal hurricane force winds observed at Hong Kong as recorded by the vessel HMS Princess Charlotte at 4 a.m. on 23 September (Table 5) revealed that the hurricane radius of the typhoon was around 70 km (which is the distance between the two observation points in Hong Kong and Macao respectively) at that time.

According to the book on historical disastrous tidal events in China (Lu Renji (陸人驥), 1984), besides Hong Kong and Macao, Typhoon 1874 also caused severe damages and flooding to various cities in western Guangdong in the vicinity of the Pearl River Estuary including Zhongshan and Panyu, and moved northwest to reach Zhaoqing, a city about 200 km west-northwest of Hong Kong.  This indicated that Typhoon 1874 moved northwest into inland of western Guangdong after passing Macao.

## 5. Reconstruction of a possible track of Typhoon 1874

Based on the analysis above, Typhoon 1874 possibly passed through the Luzon Strait at around midnight of 21 September with a maximum sustained 10-minute mean wind speed of around 130 km/h or higher near its centre and hurricane radius of over 200 km, and moved across the northeastern part of the South China Sea towards the Pearl River Estuary during the day of 22 September.  It more likely passed to the north of Dongsha in the afternoon on 22 September, skirted south of Hong Kong at around 2 a.m. on 23 September and made landfall at Macao about two hours later as more or less a super typhoon (i.e. maximum sustained 10-minute mean wind speed of 185 km/h or more) with hurricane radius of around 70 km.  It then moved northwest into inland western Guangdong.

Using the pressure observations in Hong Kong and Macao (Table 4 and Figure 5), and the Jelesnianski tropical cyclone model (Jelesnianski, 1965) shown in the equation below, given that Typhoon 1874 made landfall at Macao at 4 a.m. on 23 September with a minimum mean sea level pressure of 945 hPa near its centre, a possible track of Typhoon 1874 over the northeastern part of the South China Sea, particularly its passage along the coastal waters of eastern Guangdong and the Pearl River Estuary, can be reconstructed.

300 The Jelesnianski tropical cyclone model is described as

$$P_a(r) = \begin{cases} P_n - \dfrac{3}{4}\,(P_n - P_0)\dfrac{R}{r} & (r \geq R) \\[2em] P_0 + \dfrac{1}{4}\,(P_n - P_0)\left(\dfrac{r}{R}\right)^2 & (r < R) \end{cases}$$

where $P_a(r)$ is the mean sea level pressure at a distance r from the centre of the tropical cyclone, $P_0$ is mean sea level pressure near the centre, $P_n$ is the monthly climatological normal mean sea level pressure for the region which is taken as 1009 hPa for September in this study, R is radius of maximum winds which is defined as the distance from the centre of a tropical cyclone to the location of the 310 cyclone's maximum winds.

Table 6 shows the hourly positions, hourly minimum mean sea level pressures near the centre and the radii of maximum winds of the reconstructed possible track of Typhoon 1874 from 8 a.m. on 22 September to 11 a.m. on 23 September 1874. The atmospheric pressures at Hong Kong and Macao during the period estimated by the Jelesnianski tropical cyclone model based on the reconstructed possible track, which are also shown in Figure 5, matched rather well with the observations both at Hong Kong and Macao.

Figure 7 (a) & (b) plot the possible track of Typhoon 1874 from the Luzon Strait to inland western Guangdong reconstructed in this study. In Figure 7(a), the part of 320 the track in red was based on Table 6, the part in blue (in the morning on 22 September) was estimated by interpolation based on the qualitative analysis discussed in preceding session, and the parts in green were arbitrarily extended

to meet the requirement of input of thirteen 6-hourly positions for running the storm surge model for estimation of storm surges in Hong Kong and Macao. According to this reconstructed track, Typhoon 1874 might have moved at a high speed of about 38 km/h on average from the Luzon Strait to the Pearl River Estuary.   It picked up a northwesterly track after 10 p.m. on 22 September when it was about 190 km southeast of Hong Kong to move towards the coast along a track with more or less the same distance from Hong Kong and Macao until 2 a.m. on 23 September when it was about 60 km south-southwest of Hong Kong, the closest approach to Hong Kong.  The typhoon then took a slightly more westerly track to depart Hong Kong and moved towards Macao, resulting in a rapid rise in pressure at Hong Kong but continuing fall in pressure at Macao.   The reconstructed track also reveals the intensification and decreasing in the storm size (the hurricane radius decreased from 200 km or larger near the Luzon Strait to around 70 km when making landfall at Macao) of the typhoon during its passage across the northeastern part of the South China Sea.

## 6. Quantitative estimate of storm surge and storm tide

With the possible track reconstructed in this study, the storm surges that affected Hong Kong and Macao by Typhoon 1874 can be estimated using the storm surge model SLOSH (Sea, Lake, and Overland Surges from Hurricanes) developed by the National Oceanic and Atmospheric Administration (NOAA) of USA and being used operationally by the Hong Kong Observatory to support storm surge prediction and warning services in Hong Kong.  SLOSH requires input of thirteen 6-hourly positions, minimum mean sea level pressures near the centre and the storm sizes in terms of the radius of maximum winds along the tropical cyclone track and also uses the Jelesnianski tropical cyclone model to generate the wind and pressure fields for determining the storm surges at the point of interest (Jelesnianski *et al.*, 1992).  It was verified to have an accuracy of about 0.3 metre in root mean square error (Lee & Wong, 2007).

The 6-hourly positions, minimum mean sea level pressures near the centre and the radii of maximum winds of the reconstructed track of Typhoon 1874 in Figure 7(a) used for running SLOSH is shown in Table 7.  It can be seen that Table 7 has included the position at 10 p.m. on 22 September when Typhoon 1874 started to pick up a northwesterly track and the position at 4 a.m. on 23 September when

the typhoon made landfall at Macao. While locating maps of Hong Kong and Macao in the 1880's for obtaining the coastlines information might not be difficult, the bathymetry data with spatial resolution of about 1 km in Hong Kong and Macao waters and about 7 km in the open sea to the south of the Pearl River Estuary (the grid used for running SLOSH (Jelesnianski *et al.*, 1992)) in the 1880's would very likely be not available for running SLOSH. Instead, the earliest available digitized topographic and bathymetry information in the early 1990's were used for the estimation in this study. The maximum storm surges as estimated by SLOSH were 2.83 m (at 4:10 a.m. on 23 September) at North Point in the Victoria Harbour and 2.83 m (at 4:00 a.m. on 23 September) at Tai Po Kau in the northeastern part of Hong Kong where the tide gauges operated by the Hong Kong Observatory since the 1950s and 1960s respectively were located. In Macao, the maximum storm surge as estimated by SLOSH was 2.80 m (at 4:00 a.m. on 23 September) at Porto Interior where the tide gauge operated in Macao were located. Table 8 summarizes the estimated maximum storm surges at these three tide gauges. Locations of these three tide gauges are also marked in Figure 7 (b).

In order to estimate the extreme storm tides (storm surge on top of astronomical tide) in Hong Kong and Macao during the passage of Typhoon 1874, the astronomical tide which is caused by gravitational forcing, mostly from the Sun-Earth-Moon system, is required. For operational estimation of astronomical tide, the Hong Kong Observatory employs the harmonic method based on decade-long time series of recorded tide levels (Ip & Wai, 1990). This method is however limited in its ability to hindcast astronomical tide so long ago, as the parameters of the constituents need to be inferred from the actual tide level recorded, which were not available for that period.

Instead of adopting the harmonic method direct, the astronomical tide during the passage of Typhoon 1874 was estimated by analogue with dates of similar Sun-Earth-Moon configuration in this study. Astronomical configurations could be found through referencing the Multiyear Interactive Computer Almanac (MICA) created by the U.S. Naval Observatory (USNO), which utilizes the Jet Propulsion Laboratory (JPL) DE405 ephemerides for position calculations of the Sun, Moon and major planets. The geometric geocentric positions (equator of J2000.0) of the Sun and the Moon during 22-23 September 1874 were computed with MICA and, after searching through dates since the 1950's (the earliest when tide level observations were available), it was found that 22-23 September 1950, 22-23

September 1969, 22-23 September 1988 and 22-23 September 2007 had the higher relevance to 22-23 September 1874, which is not surprising as 1950, 1969, 1988 and 2007 were exactly four Metonic cycles, each of 19 years, after 1874.

Using the earliest available 19-year tide level observations at North Point (1954 to 1972), Tai Po Kau (1969 to 1987) and Porto Interior (1998 to 2016), the astronomical tide during 22-23 September 1874 at the three tide gauge stations could then be taken as the astronomical tide during 22-23 September 1950, 22-23 September 1969 and 22-23 September 2007 respectively.

400 Combining the estimated storm surges and hindcasted astronomical tides, the peak storm tides which are also shown in Table 8 were estimated to be 4.88 m above Hong Kong Chart Datum (at 4:10 a.m. on 23 September) at North Point, 4.95 m above Hong Kong Chart Datum (at 4:00 a.m. on 23 September) at Tai Po Kau and 5.37 m above Macao Chart Datum (at 4:00 a.m. on 23 September) at Porto Interior in Macao. Figure 8, 9 and 10 plot the time series of the hindcasted astronomical tides, estimated storm surges and storm tides at North Point (Hong Kong), Tai Po Kau (Hong Kong) and Porto Interior (Macao) respectively from 11 a.m. on 22 September to 6 p.m. on 23 September 1874.

It can be seen that the hindcasted astronomical tide at North Point showed a low
410 tide from midnight to around 2 a.m. on 23 September which was consistent with what the Harbour Master mentioned in his report in the Hong Kong Government Gazette of 17 October 1874 (The Hong Kong Government, 1874) – "Although, according to ordinary calculation it should have been low water at two o'clock". Furthermore, given that the observations in the historical documents were taken by human eyes at night time and not at the locations of the respective tide gauge stations, (a) the difference between the estimated storm tide of 3.69 m at 3 a.m. and the hindcasted astronomical high tide of 2.28 m at around 6:30 a.m. on 23 September at North Point (which was 1.41 m) was slightly lower than but comparable with the qualitative description of the rise in sea levels as recorded in
420 the report of the Harbour Master in the Hong Kong Government Gazette of 17 October 1874 (The Hong Kong Government, 1874) – "By three, the water had risen to from five to six feet (equivalent to 1.52 m to 1.83 m) above its high water level" (meaning that the storm tide was 1.52 m to 1.83 m above the astronomical high tide in Hong Kong), and (b) the difference between the estimated maximum storm tide of 5.37 m at 4 a.m. and the hindcasted astronomical high tide of 2.77

m at around 6 a.m. on 23 September at Porto Interior in Macao (which was 2.60 m) was slightly higher than but still considered comparable with the qualitative description of rise in sea levels as recorded in the publication "O Maior Tufao De Macau - 22 e 23 de Setembro de 1874" by Father Manuel Teixeira (1974) – "storm surge which caused severe flooding of up to 7 feet (equivalent to 2.13 m) above the high tide level" (meaning that the maximum storm tide was up to 2.13 m above the astronomical high tide in Macao).

## 7. Results and Discussions

Analysing the available weather records in historical documents, a possible track of Typhoon 1874 as shown in Figure 7(a) and Figure 7(b) was reconstructed in this study.  It has to be noted that the parts of the reconstructed track over the western North Pacific and southwestern part of China (plotted in green in Figure 7(a)) were arbitrarily extended to meet the requirement of input of thirteen 6-hourly positions for running the storm surge model for estimation of storm surges in Hong Kong and Macao.  The limited weather observations in the Luzon Strait area, though not sufficient enough to enable a detailed estimation of the positions of Typhoon 1874 moving over the Luzon Strait (plotted in blue in Figure 7(a)), have provided evidence that the typhoon had likely moved across the Luzon Strait between Vigan and Batan with typhoon intensity on the early morning of 22 September.  For the reconstructed track over the northeastern part of the South China Sea, the Pearl River Estuary and western Guangdong (plotted in red in Figure 7(a) and the whole track in Figure 7(b)), a more quantitative and reliable estimation of the hourly positions, hourly minimum mean sea level pressures near the centre and the radii of maximum wind became possible by using the Jelesnianski tropical cyclone model based on the more comprehensive weather observations taken in Hong Kong and Macao.  Besides reproducing the trends of change of atmospheric pressure with time at Hong Kong and Macao, including the rapid fall of atmospheric pressure at Macao from 2 a.m. to 4 a.m. on 23 September while the atmospheric pressure at Hong Kong was rising (Figure 5), the atmospheric pressure readings taken at Hong Kong and Macao could also be reproduced well.  Comparing the hourly atmospheric pressures at Hong Kong and Macao during the period estimated by the Jelesnianski tropical cyclone model based on the hourly positions, hourly minimum mean sea level pressures near the centre and the hourly radii of maximum winds of Typhoon 1874 in Table 6 with

the corresponding available hourly atmospheric pressure observations taken by the Hong Kong Harbour Master Office and Vessel HMS Princess Charlotte in Hong Kong (where the pressure readings were taken near mean sea level and closest to the Hong Kong Observatory) and Gunboat Tejo in Macao (where the pressure readings were taken near mean sea level and at Porto Interior), the root-mean-square of the differences were 4.0 hPa, 4.6 hPa and 2.7 hPa respectively.  The differences were even smaller for the period from 8 p.m. on 22 September (when the storm surge at North Point started to rise and before the typhoon picked up a northwesterly track) to 4 a.m. on 23 September (when the storm surges at North Point, Tai Po Kau and Porto Interior were almost at the highest and the typhoon had made landfall at Macao) with root-mean-squares of difference of 2.7 hPa, 3.1 hPa and 1.7 hPa respectively.  Furthermore, the reconstructed track also matched well with the observed wind direction changes at Hong Kong reported by the Harbour Master and HMS Princess Charlotte as shown in Table 5 during the approach and departure of the typhoon.  Combining Figure 6 and Figure 7 could reveal that the wind direction at Hong Kong would veer gradually from northwesterly to northeasterly during the day on 22 September, and continue to veer to easterly and then southeasterly during the evening on 22 September and early morning of 23 September.  Such sequence of wind direction change would not occur if the typhoon approached Hong Kong from the southeast or south during the day on 22 September.

The quantitative weather observations also helped reveal some special characteristics of the typhoon.  Besides a fast moving typhoon in the northeastern part of the South China Sea (around 38 km/hour from Luzon Strait to the Pearl River Estuary), the observations suggested that Typhoon 1874 had undergone rapid intensification as well as decrease in storm size to become a more intense and compact storm in several hours before making landfall at Macao.  The minimum mean sea level pressure near the centre of the Typhoon 1874 decreased from 955 hPa at 11 p.m. on 22 September to 945 hPa at 4 a.m. on 23 September and its radius of maximum winds decreased from 55 km to 25 km during the same period (Table 6).  Such rapid intensification before making landfall at the south China coast was not uncommon.  Recent examples were Severe Typhoon Vincente in 2012 (HKO, 2013) and Super Typhoon Hato in 2017 (https://www.weather.gov.hk/informtc/hato17/hato.htm; Lau & Chan, 2017).

On the other hand, the descriptive weather phenomena in historical documents together with the quantitative weather observations in Hong Kong could provide information for a rough estimation of the path of the typhoon over the northeastern part of the South China Sea, a better estimation of this part of the track would have been possible if the logbooks of the three ships (namely British ship Onward, American ship Highlander, German ship Amanda) when they were near Dongsha during the passage Typhoon 1874 could be found.

Overall speaking, the quantitative weather observations near Luzon and the Pearl River Estuary (Hong Kong and Macao) were very useful for estimating a reasonable track of the typhoon when it passed through the Luzon Strait, the northeastern part of the South China Sea and the Pearl River Estuary. The study results demonstrated the usefulness of weather observations in historical documents and the importance and value of the international joint effort on climatological data rescue and retrieval of the historical climate data to studies of historical weather events.

According to the reconstructed track, Typhoon 1874 resembled the tracks of the typhoons in Table 1 which brought severe storm surges to Hong Kong. The reconstructed track of the typhoon itself can be used as a possible scenario for assessment of the present and future storm surge risk in the Pearl River Estuary together with the other historical typhoons.

This study also estimated the storm surges and storm tides at North Point and Tai Po Kau in Hong Kong and Port Interior in Macao brought by Typhoon 1874 by running SLOSH using the reconstructed track. It can be seen that both the estimated maximum storm surge and storm tide at North Point in the Victoria Harbour (shown in Table 8) were higher than those brought by the typhoons in Table 1, i.e. higher than those experienced since the establishment of the Hong Kong Observatory in 1883, including the recent records set by super typhoon Mangkhut on 16 September 2018 (maximum storm surge of 2.35 m and storm tide of 3.88 m above Hong Kong Chart Datum at Quarry Bay in the Victoria Harbour). Such an extreme sea level which would be probable in the history of Hong Kong according to this study revealed that the risk assessment on extreme sea level in Hong Kong based on the available instrumental records (since the 1950's) or even all available records after the establishment of the Hong Kong Observatory (since 1883) might be on the optimistic side. A more detailed

frequency analysis of extreme sea levels taking Typhoon 1874 as well as other historical significant storm surge events such as the typhoons in 1906, 1936 and 1937 in Table 1 into account is essential for a more realistic storm surge risk assessment for Hong Kong.

However, it should be noted that, besides the uncertainty of the typhoon track (position, intensity and storm size in terms of radius of maximum winds) and the uncertainty of the estimation by SLOSH, the estimated storm surges in Hong Kong and Macao were also subjected to the uncertainty of the change of the local bathymetry and coastline since 1874.   As the bathymetry and coastline of the Pearl River Estuary, including Hong Kong and Macao in 1874 were not readily available, the earliest readily available digitized bathymetry and coastline of the region in the early 1990's were used for running SLOSH in this study.  Due to the rapid development of the region in the past 100 years or so, the bathymetry and coastline of the region should have changed quite a lot and would cause a certain degree of uncertainty to the estimated storm surges.  To show the sensitivity of the changes in topography and bathymetry, a comparison of the SLOSH results of Typhoon 1874 using topography and bathymetry data in the 1990's and 2010's was conducted.  The results showed that the maximum storm surges at North Point, Tai Po Kau and Macao using topography and bathymetry data in the 1990's (2010's) were 2.83 m (2.71 m), 2.83 m (2.77 m) and 2.80 m (2.68 m) respectively.  Despite significant coastal development had occurred such as reclamations and building of new airports from the 1990's to 2010's, the differences of the estimated maximum storm surges at these three tide gauge stations generated by Typhoon 1874 were less than 0.12 m, well within the accuracy of SLOSH of about 0.3 m in root mean square error (Lee & Wong, 2007).

Furthermore, the difference between the mean sea levels in 1874 and those in the years of astronomical tides used in this study for estimating the storm tides (1950 for North Point, 1969 for Tai Po Kau, 2007 for Porto Interior in Macao) could also bring some uncertainties to the storm tides estimated in this study. According to the 5th Assessment Report of the Intergovernmental Panel on Climate Change (IPCC-AR5), global average sea level rose at 1.7 mm per year during the period 1901-2010 (Church *et al*., 2013).  Assuming a similar rate of sea level change at the Pearl River Estuary, the difference in the mean sea levels could roughly cause an additional 0.1 to 0.2 m to the storm tides estimated in this study.

Given the above uncertainties, care has to be taken when comparing the storm surges and storm tides of Typhoon 1874 estimated in this study with the observed storm surges and storm tides brought by the other historical typhoons.

**Data Availability**

All historical data under "Section 3. Historical documents available for the study" are available in public domain. The storm surge and storm tide data generated by SLOSH as well as the astronomical tide data estimated in this study are available at the Hong Kong Observatory on request.

**Acknowledgements**

The authors are grateful to Dr Rob Allan and Dr Clive Wilkinson for their help to locate and provide the invaluable historical documents for this study. The authors also expressed appreciation to comments provided by Messrs CM Shun and ST Chan on the draft manuscript.

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

Table 1. Records of major storm surges recorded in the Victoria Harbour in Hong Kong since the establishment of the Hong Kong Observatory in 1883

| Typhoon name | Year | Maximum storm surge at Victoria Harbour (above astronomical tide) (m) | Maximum storm tide at Victoria Harbour (above Chart Datum) (m) | Number of deaths |
|---|---|---|---|---|
| - | 1906 | 1.83[1] | 3.35[1,2] | over 10,000[3] |
| - | 1936 | 1.92[1] | 3.81[1] | 20 |
| - | 1937 | 1.98[1] | 4.05[1] | ~11,000[3] |
| Wanda | 1962 | 1.77 | 3.96[4] | 130 deaths 53 missing |
| Hope | 1979 | 1.45 | 2.78[5,6] | 12 |
| Hagupit | 2008 | 1.43 | 3.53[6] | 0 |
| Hato | 2017 | 1.18 | 3.57[6] | 0 |
| Mangkhut | 2018 | 2.35 | 3.88[6] | 0 |

630

[1] Based on tide pole observations, field surveys or reports of local residents. The operation of tide gauge network started in 1952.

[2] Information on the reference level for the storm tide reading was not available as the Chart Datum was not yet established in 1906.

[3] According to press reports.

[4] Recorded at the tide gauge station at North Point in the Victoria Harbour

[5] The maximum storm surge did not occur at astronomical high tide.

[6] Recorded at the tide gauge station at Quarry Bay (about 500 m east of North Point) in the Victoria Harbour

Table 2. Chronological summary of weather observations at Vigan (extracted from Annex I) during the passage of Typhoon 1874

| Date/Local Time | Atmospheric Pressure (hPa) | Winds | Weather |
|---|---|---|---|
| 21 September/ dawn | 993.9 | Calm | Rainy |
| 21 September/ 4 p.m. | 988.6 | Slight SW | Rain aplenty |
| 21 September/ midnight | 993.3 | Hurricane W | - |
| 22 September/ 8 a.m. | 996.9 | SSW becoming S | Much rain |

Note: the following conversion factor has been adopted:

1 mm mercury = 1.33322387 hPa
Reference: Smithsonian Meteorological Tables, 4th revised edition, 1918

Table 3. Chronological summary of weather observations at Batan Islands (extracted from Annex I) during the passage of Typhoon 1874

| Date/Local Time | Atmospheric Pressure (hPa) | Winds |
|---|---|---|
| 21 September/ twilight | Began to descend conspicuously | - |
| 21 September/ 8 p.m. | Rapid fall | Strong NNE |
| 21 September/ 10 p.m. | - | Hurricane |
| 21 September/ 11 p.m. | 982.6 | NNE with frightful violence |
| 22 September/ 1 a.m. | 965.3 | Maximum intensity E |
| 22 September/ 4 a.m. | Rapid rise since 1 a.m. | SE |

650    Note: the following conversion factor has been adopted:

1 mm mercury = 1.33322387 hPa
Reference: Smithsonian Meteorological Tables, 4th revised edition, 1918

# Table 4. Reports of pressure observations in Hong Kong and Macao during the passage of Typhoon 1874

| Date/Time | Hong Kong Police Central Station barometer readings[1] | Hong Kong Harbour Master office barometer readings reduced to sea level[2] | Messrs. Geo Falconer & Co.'s, Hong Kong aneroid barometer readings[3] | Messrs Geo Falconer & Co.'s, Hong Kong mercurial barometer readings[3] | HMS Princess Charlotte, mooring off Kowloon[4] | Gunboat Tejo at inner harbor of Macao[5] | Macao pressure readings[6] |
|---|---|---|---|---|---|---|---|
| Local time | inches Hg (hPa) | inches Hg (hPa) | inches Hg (hPa) | inches Hg (hPa) | inches Hg (hPa) | mm Hg (hPa) | inches Hg (hPa) |
| 22/0900 | | | | 29.852(1010.9) | | | |
| 22/morning | | 29.85 (1010.8) | | | | | |
| 22/1100 | | | | | | | 29.59 (1002.0) |
| 22/1200 | | | | | 29.78 (1008.5) | 755.382 (1007.1) | |
| 22/1300 | | | | 29.770 (1008.1) | | | |
| 22/1400 | | | | | | | |
| 22/1500 | | | | | | 753.604 (1004.7) | |
| 22/1600 | | 29.74 (1007.1) | | | | | 29.55 (1000.7) |
| 22/1700 | | | | 29.704 (1005.9) | | | 29.54 (1000.3) |
| 22/1800 | | | | | | 751.826 (1002.4) | 29.53 (1000.0) |
| 22/1900 | | | | 29.634 (1003.5) | | 751.318 (1001.7) | 29.50 (999.0) |
| 22/1930 | 29.50 (999.0) | | | | | | |
| 22/1945 | | 29.63 (1003.4) | | | | | |
| 22/2000 | | | | 29.602 (1002.4) | | 750.810 (1001.0) | 29.48 (998.3) |
| 22/2100 | 29.40 (995.6) | | 29.500 (999.0) | | 29.55 (1000.7) | 750.302 (1000.3) | 29.44 (997.0) |

| | | | | | | | |
|---|---|---|---|---|---|---|---|
| 22/2115 | | 29.57 (1001.4) | | | | | |
| 22/2125 | | | | | | | |
| 22/2200 | 29.30 (992.2) | | 29.408 (995.9) | 29.500 (999.0) | 29.46 (997.6) | 748.779 (998.3) | 29.38 (994.9) |
| 22/2215 | | 29.45 (997.3) | | | | | |
| 22/2230 | | 29.40 (995.6) | | | | | |
| 22/2300 | | | 29.325 (993.1) | 29.345 (993.7) | 29.27 (991.2) | 746.493 (995.2) | 29.33 (993.2) |
| 22/2330 | 29.20 (988.8) | | | | | | 29.25 (990.5) |
| 22/2400 | 29.15 (987.1) | | 29.100 (985.4) | 29.200 (988.8) | 29.18 (988.1) | 743.699 (991.5) | 29.21 (989.2) |
| 23/0005 | 29.10 (985.4) | | | | | | |
| 23/0015 | 29.06 (984.1) | | | | | | |
| 23/0025 | 29.02 (982.7) | | | | | | |
| 23/0030 | | 29.12 (986.1) | | | | | 29.18 (988.2) |
| 23/0040 | 28.95 (980.4) | | | | | | |
| 23/0045 | 28.95 (980.4) | | | | | | |
| 23/0056 | 28.94 (980.0) | | | | | | |
| 23/0100 | | 29.05 (983.7) | 28.897 (978.6) | 28.950 (980.4) | 28.87 (977.7) | 738.619 (984.7) | 29.05 (983.7) |
| 23/0103 | 28.91 (979.0) | | | | | | |
| 23/0107 | 28.90 (978.7) | | | | | | |
| 23/0110 | 28.90 (978.7) | | | | | | |
| 23/0112 | | 29.00 (982.1) | | | | | |
| 23/0120 | 28.88 (978.0) | | | | | | |
| 23/0123 | 28.85 (977.0) | | | | | | |

| | | | | | | | |
|---|---|---|---|---|---|---|---|
| 23/0127 | 28.84 (976.6) | | | | | | |
| 23/0130 | | 28.95 (980.4) | | | | | 28.89 (978.3) |
| 23/0134 | 28.82 (976.0) | | | | | | |
| 23/0135 | 28.81 (975.6) | | | | | | |
| 23/0137 | 28.80 (975.3) | | | | | | |
| 23/0143 | 28.78 (974.6) | | | | | | |
| 23/0148 | 28.76 (973.9) | | | | | | |
| 23/0155 | 28.75 (973.6) | | | | | | |
| 23/0200 | | 28.88 (978.0) | 28.760 (973.9) | 28.870 (977.7) | 28.83 (976.3) | 731.761 (975.6) | 28.8 (975.3) |
| 23/0203 | 28.73 (972.9) | | | | | | |
| 23/0215 | 28.77 (974.3) | | 28.727 (972.8) | 28.752 (973.7) | | | |
| 23/0220 | 28.80 (975.3) | | | | | | |
| 23/0230 | | | 28.785 (974.8) | 28.810 (975.6) | | | 28.61 (968.8) |
| 23/0240 | 28.85 (977.0) | | | | | | |
| 23/0245 | 28.90 (978.7) | | | | | | |
| 23/0250 | 28.90 (978.7) | | | | | | |
| 23/0255 | 28.95 (980.4) | | | | | | |
| 23/0300 | 28.97 (981.0) | 29.04 (983.4) | 28.990 (981.7) | 29.235 (990.0) | 29.12 (986.1) | 715.759 (954.3) | 28.22 (955.6) |
| 23/0305 | 29.00 (982.1) | | | | | | |
| 23/0310 | 29.05 (983.7) | | | | | | |
| 23/0320 | 29.10 (985.4) | | | | | | |
| 23/0330 | 29.15 (987.1) | | 29.184 (988.3) | 29.286 (991.7) | | | 28.11 (951.9) |

| | | | | | | | |
|---|---|---|---|---|---|---|---|
| 23/0335 | 29.20 (988.8) | 29.22 (989.5) | | | | | |
| 23/0345 | 29.25 (990.5) | | | | | | |
| 23/0400 | 29.28 (991.5) | | 29.320 (992.9) | 29.315 (992.7) | 29.40 (995.6) | 709.155 (945.5) | 27.94 (946.2) |
| 23/0405 | 29.30 (992.2) | | | | | | |
| 23/0410 | | 29.32 (992.9) | | | | | |
| 23/0415 | | | | | | | 28.06 (950.2) |
| 23/0420 | 29.34 (993.6) | | | | | | |
| 23/0430 | | | | | | | 28.23 (956.0) |
| 23/0445 | | | | | | | 28.40 (961.7) |
| 23/0500 | | 29.49 (998.6) | | | 29.53 (1000.0) | 724.809 (966.3) | 28.59 (968.2) |
| 23/0515 | | | | | | | 28.77 (974.3) |
| 23/0530 | | | | | | | 28.98 (981.4) |
| 23/0540 | | 29.58 (1001.7) | | | | | |
| 23/0545 | | | | | | | 29.08 (984.8) |
| 23/0600 | | | | | 29.65 (1004.1) | 744.461 (992.5) | 29.16 (987.5) |
| 23/0615 | | | | | | | 29.25 (990.5) |
| 23/0630 | | | | | | | 29.28 (991.5) |
| 23/0700 | | | | | 29.72 (1006.4) | 749.540 (999.3) | 29.33 (993.2) |
| 23/0730 | | | | | | | 29.42 (996.3) |
| 23/0800 | | | | | 29.79 (1008.8) | 753.604 (1004.7) | 29.47 (998.0) |
| 23/0815 | | 29.80 (1009.1) | | | | | |
| 23/0830 | | | | | | | 29.52 (999.7) |

| Time | | | | | | | |
|---|---|---|---|---|---|---|---|
| 23/0900 | | | | | 29.82 (1009.8) | 755.382 (1007.1) | 29.55 (1000.7) |
| 23/0930 | | 29.84 (1010.5) | | | | | |
| 23/1000 | | | | | 29.87 (1011.5) | | 29.58 (1001.7) |
| 23/1200 | | | | | | 756.144 (1008.1) | |

1 Published in the Hong Kong Government Gazette of 17 Oct 1874.
2 Published in the Hong Kong Government Gazette of 17 Oct 1874 (readings for 22/morning, 22/1600, 22/2230 and 23/0200) and The China Mail of 23 Sep 1874 (others).
3 Published in the China Mail of 23 Sep 1874.
4 Logbook of HMS Princess Charlotte.
5 Observations of Gunboat Tejo published in the Bulletin of the Macao Province and Timor Anno, 1874-Vol. XX- No. 41 Saturday 10 October.
6 Report of Port Captain of Macao published in the Bulletin of the Macao Province and Timor Anno, 1874-Vol. XX- No. 44 Saturday 31 October.

Note: the following conversion factors have been adopted:

(i)  1 inches of mercury = 33.86395 hPa
(ii) 1 mm mercury = 1.33322387 hPa
Reference: Smithsonian Meteorological Tables, 4th revised edition, 1918

## Table 5. Reports of winds in Hong Kong and Macao during the passage of Typhoon of 1874

| Date/Time | Wind reports in Hong Kong Police Central Station[1] | | Wind reports in Hong Kong Harbour[1] | | Wind reports by HMS Princess Charlotte[2]* | | Wind reports in Macao[3] | |
|---|---|---|---|---|---|---|---|---|
| Local time | Direction | Beaufort Force | Direction | Beaufort Force | Direction | Beaufort Force | Direction | Beaufort Force |
| 22/morning | | | NW | | | | | |
| 22/1200 | | | | | N | 4 | N / NNE | 4 |
| 22/1500 | | | | | | | N | 3 |
| 22/1600 | | | NNW | Fitful gusts | N | 6 to 7 | | |
| 22/1800 | | | | | | | NW | 5 |
| 22/1900 | | | | | | | NW | 5 |
| 22/2000 | | | | | N | 4 to 6 | | |
| 22/2100 | | | | | | | NNW | 6 |
| 22/2200 | Gusts heavy and frequent. | | | | | | NNW | 7 |
| 22/2230 | | | N | | | | | |
| 22/2300 | | | | | | | NNW | 8 |
| 22/2330 | Shifting to 70. | | | | | | | |
| 22/2400 | | | | | NNE | 8 to 10 | NNW | 10 |

| Time | | | | | | | | |
|------|---|---|---|---|---|---|---|---|
| 23/0045 | Gusts lighter and long | | | | | | | |
| 23/0100 | | | | | NE | 8 to 11 | NNW, N | 10 |
| 23/0107 | Gusts heavier and continuous. | | | | | | | |
| 23/0123 | Gusts heavy but longer intervals. | | | | | | | |
| 23/0200 | | | Suddenly shifted to NE and then to ENE | Terrible violence | ENE | 9 to 12 | N | 11 |
| 23/0300 | | | | | ESE | 10 to 12 | N | 12 |
| 23/0400 | | | | | ESE | 10 to 12 | NE, ENE, NW, E, SE | 12 |
| 23/0405 | Gradually veered to SE | | | | | | | |
| 23/0500 | | | | | SE | 7 to 9 | SE | 12 |
| 23/0600 | | | | | SE | 7 to 9 | SSE | 11 |
| 23/0700 | | | | | SSE | 7 to 8 | SSE | 11 |
| 23/0800 | | | | | SSE | 7 to 8 | S | 9 |
| 23/0900 | | | | | SSE | 7 to 8 | S | 6 |
| 23/1200 | | | | | SSE | 3 to 6 | S | 4 |

* There were uncertainties in reading some of the hand-written characters.

1 Published in the Hong Kong Government Gazette of 17 Oct 1874.
2 Logbook of HMS Princess Charlotte.
3 Observations of Gunboat Tejo published in the Bulletin of the Macao Province and Timor Anno, 1874-Vol. XX- No. 41 Saturday 10 October.

Table 6. Hourly positions, hourly minimum mean sea level pressures near the centre and the radii of maximum winds of the reconstructed possible track of Typhoon 1874 from 8 a.m. on 22 September to 11 a.m. on 23 September 1874.

| DDHH (Local Time) | Latitude (°N) | Longitude (°E) | Pressure near centre (hPa) | Radius of maximum winds (km) |
|---|---|---|---|---|
| 2208 | 21.1 | 119.1 | 965 | 60 |
| 2209 | 21.1 | 118.8 | 965 | 60 |
| 2210 | 21.1 | 118.5 | 965 | 60 |
| 2211 | 21.1 | 118.2 | 965 | 60 |
| 2212 | 21.1 | 117.9 | 965 | 60 |
| 2213 | 21.1 | 117.6 | 965 | 60 |
| 2214 | 21.1 | 117.3 | 965 | 60 |
| 2215 | 21.1 | 117.0 | 965 | 60 |
| 2216 | 21.1 | 116.7 | 960 | 60 |
| 2217 | 21.1 | 116.4 | 960 | 60 |
| 2218 | 21.1 | 116.1 | 960 | 60 |
| 2219 | 21.1 | 115.8 | 960 | 60 |
| 2220 | 21.1 | 115.5 | 960 | 55 |
| 2221 | 21.1 | 115.2 | 960 | 55 |
| 2222 | 21.1 | 114.9 | 955 | 55 |
| 2223 | 21.2 | 114.6 | 955 | 55 |
| 2300 | 21.4 | 114.4 | 950 | 55 |
| 2301 | 21.5 | 114.2 | 950 | 55 |
| 2302 | 21.8 | 114.0 | 950 | 45 |
| 2303 | 22.0 | 113.7 | 950 | 35 |
| 2304 | 22.2 | 113.5 | 945 | 25 |
| 2305 | 22.2 | 113.1 | 955 | 35 |
| 2306 | 22.3 | 112.6 | 960 | 40 |
| 2307 | 22.4 | 112.2 | 965 | 45 |
| 2308 | 22.6 | 111.9 | 970 | 45 |
| 2309 | 22.8 | 111.5 | 975 | 45 |
| 2310 | 23.0 | 111.1 | 980 | 45 |
| 2311 | 23.0 | 110.6 | 985 | 45 |

Table 7. The 13 points of the 6-hourly positions, minimum mean sea level pressures near the centre and the radii of maximum winds of the reconstructed possible track of Typhoon 1874 used for running SLOSH.

| MMDD/HH (Local Time) | Latitude (°N ) | Longitude (°E ) | Pressure near centre (hPa) | Radius of maximum winds (km) |
|---|---|---|---|---|
| 0921/04 | 18.0 | 131.5 | 965 | 80 |
| 0921/10 | 18.5 | 128.3 | 965 | 80 |
| 0921/16 | 19.2 | 125.5 | 965 | 80 |
| 0921/22 | 20.0 | 123.1 | 965 | 80 |
| 0922/04 | 20.7 | 120.5 | 965 | 70 |
| 0922/10 | 21.1 | 118.5 | 965 | 60 |
| 0922/16 | 21.1 | 116.7 | 960 | 60 |
| 0922/22 | 21.1 | 114.9 | 955 | 55 |
| 0923/04 | 22.2 | 113.5 | 945 | 25 |
| 0923/10 | 23.6 | 111.8 | 980 | 45 |
| 0923/16 | 24.5 | 109.4 | 985 | 60 |
| 0923/22 | 25.1 | 107.1 | 990 | 60 |
| 0924/04 | 26.0 | 105.0 | 995 | 60 |

Table 8. The estimated maximum storm surges and maximum storm tides at North Point and Tai Po Kau in Hong Kong and Port Interior in Macao during the passage of Typhoon 1874

| Location | Maximum Storm Surge (m) | Maximum Storm Tide (m) |
| --- | --- | --- |
| North Point (Hong Kong) | 2.83 | 4.88[1] |
| Tai Po Kau (Hong Kong) | 2.83 | 4.95[1] |
| Porto Interior (Macao) | 2.80 | 5.37[2] |

1 Above Hong Kong Chart Datum
2 Above Macao Chart Datum

**Extract from The Selga Chronology (Part I: 1348-1900) with information on the 1874 typhoon (R. García-Herrera, P. Ribera, E. Hernández, L. Gimeno:'Typhoons in the Philippine Islands 1566-1900'. Submitted to Journal of Climate.)**

The dawn of the 21st in Vigan was cloudy and rainy. The barometer registered 745.5 mm with calm. At four o'clock in the afternoon, it began to rain aplenty. The barometer registered 741.5 mm with a slight wind from the SW which continued until midnight, when it veered to the W and reached hurricane force. The barometer remained at 745 mm until 8 in the morning when the wind backed to SSW and to the S with much rain; the barometer at 747.7 mm. At midday perfect calm; and the barometer began to rise; in the afternoon it cleared up with a slight wind from the S. Jose Serra described in this guise the effects of the storm which burst over Batan Islands on the 22nd of September, 1874: "At twilight on the 21st of September, the barometer began to descend in a conspicuous manner. At 8 at night, a strong NNE wind began to blow, while the barometer continued its rapid fall. At 10, a hurricane of extraordinary intensity was present. The aneroid barometers registered 737.0 mm ay 11, the wind veered to the NNE with frightful violence, while the barometer continued its descent. From this hour until 1, the wind blew with such a terrifying violence that the aneroides dropped 0.20, until they registered 724.0 mm. At the said hour, the wind veered to the E, where it acquired its maximum intensity, and produced the greatest havoc. From this hour, the barometer began to rise as fast as it had fallen. At 4, the SE wind began to blow, and at 5, I was able to go out and visit the town. The typhoon was over, but it left in its trails desolation and death. This island, being very small, and very near to Sabtang and Isbat the typhoon was experienced in all these places with the same intensity as in the capital. The churches and mission houses received damages, especially on the roofs. The church of the town of San Carlos was completely destroyed. The tribunals and schools were left in pitiable state. Seven houses made of lime and stone in the capital collapsed. One hundred sixty-four houses of wood and cogon were destroyed. It should not escape notice that while in Luzon houses were made earthquake proof, the natives in these isles make their cottages baguio proof, so that the devastating effects of the typhoons in these islands will ever be as great as in any town of Luzon. The winds of this storm were from the 1st and 2nd quadrant.

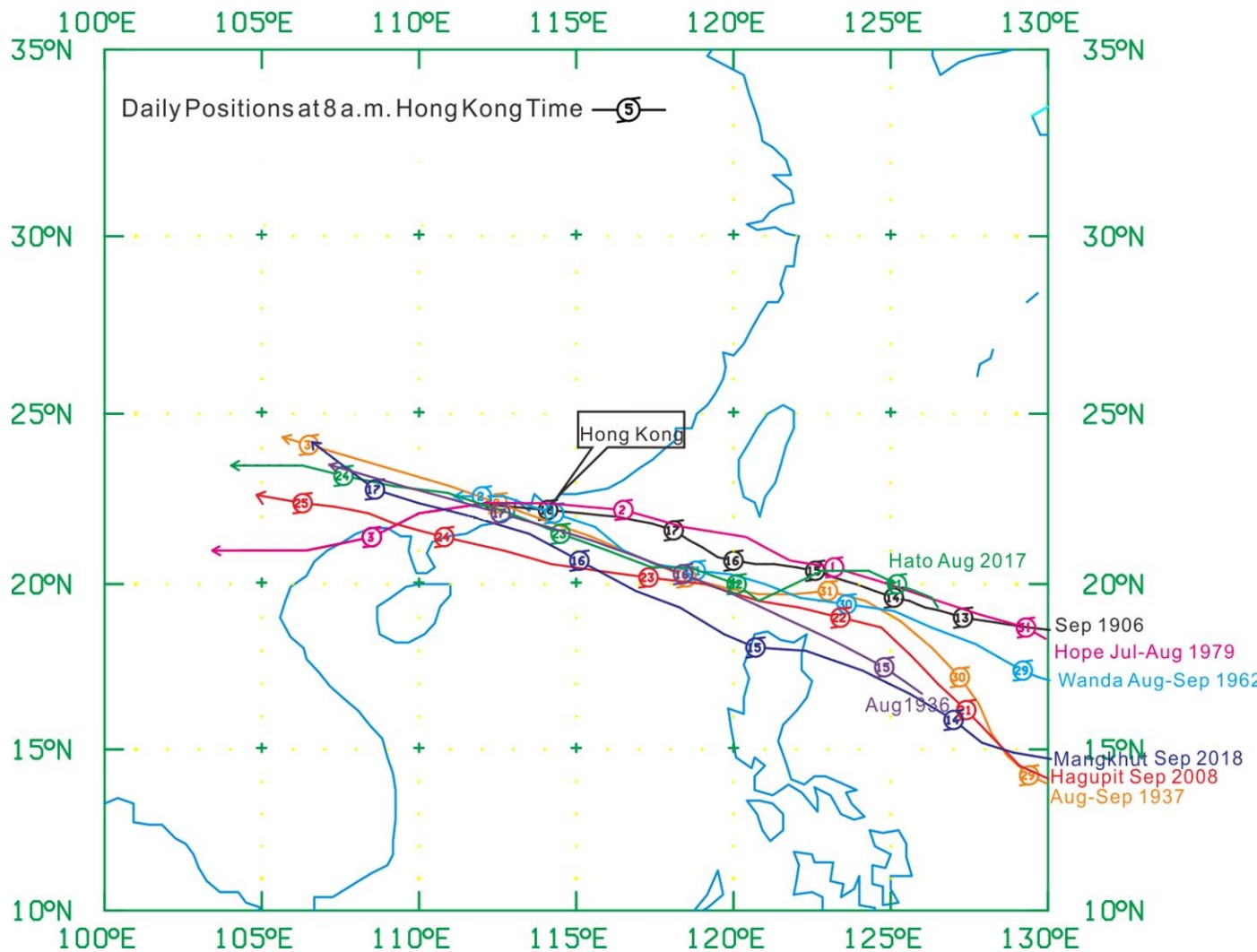

Figure 1. Daily positions at 8 a.m. Hong Kong Time of the typhoons affecting Hong Kong on 13-18 September 1906 (Typhoon Sep 1906), 15-17 August 1936 (Typhoon Aug 1936), 29 August to 3 September 1937 (Typhoon Aug-Sep 1937), Super Typhoon Wanda from 29 August to 2 September 1962 (Wanda Aug-Sep 1962), Super Typhoon Hope from 31 July to 3 August 1979 (Hope Jul-Aug 1979), Severe Typhoon Hagupit on 21-25 September 2008 (Hagupit Sep 2008), Super Typhoon Hato on 21-24 August 2017 (Hato Aug 2017) and Super Typhoon Mangkhut on 14-17 September 2018 (Mangkhut Sep 2018).

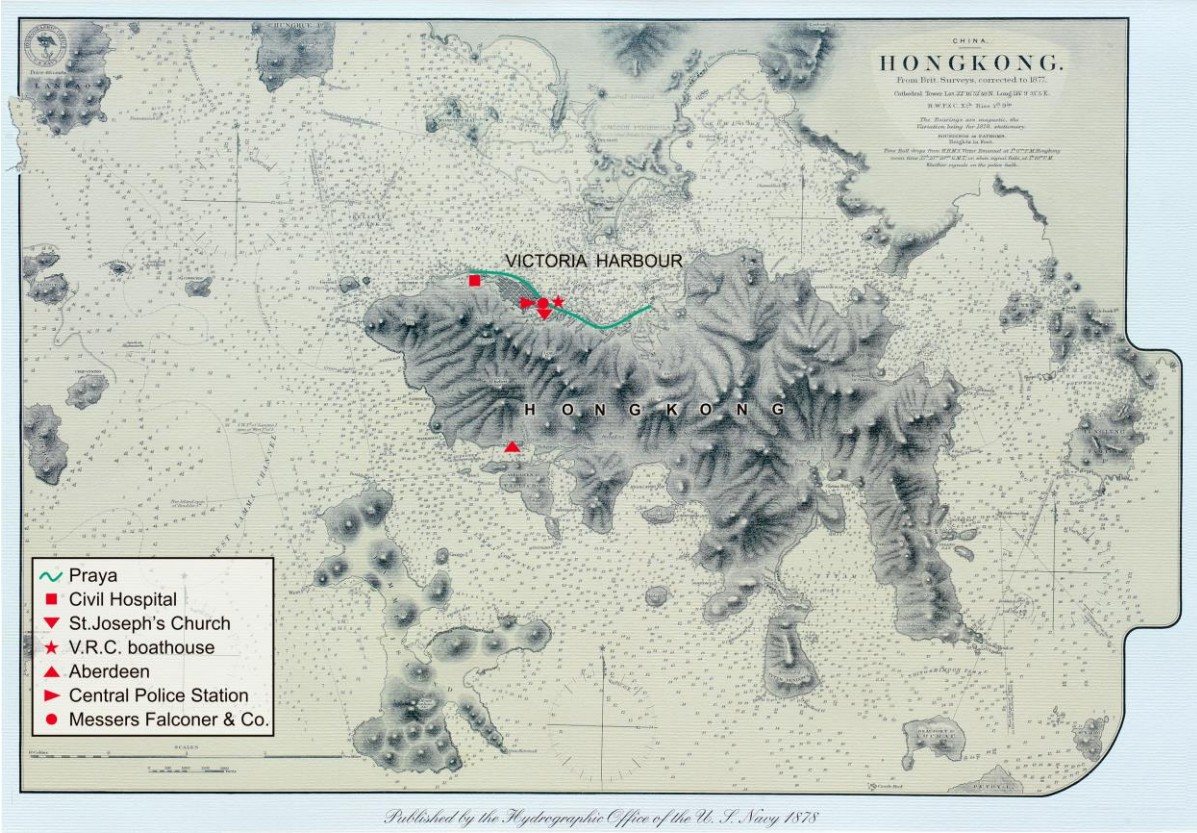

Figure 2. Historical Hong Kong map published by the Hydrographic Office of the U.K. Navy in 1878 (Courtesy of Mr. C. M. Shun), showing the Praya along the northern coast of the Hong Kong Island (shown in 〜) and the locations of Civil Hospital (■), St. Joseph's Church (▼), V.R.C. boathouse (★), Aberdeen (▲), Central Police Station (▶) and Messers Falconer & Co (●) in the Hong Kong Island.

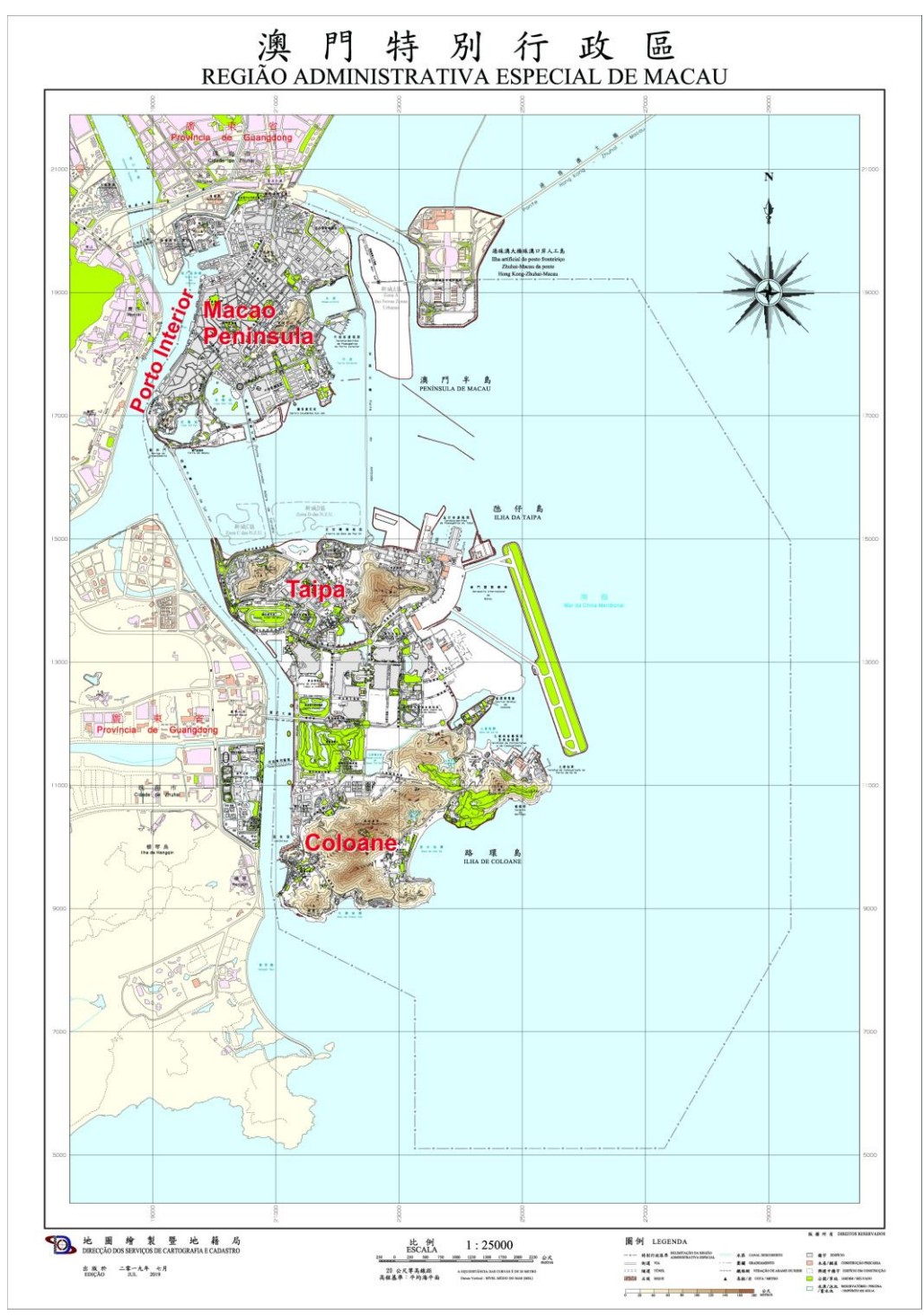

Figure 3. A map of Macao in 2019 showing the locations of Porto Interior, Macao Peninsula, Taipa and Coloane (Courtesy of Macao Special Administrative Region Government - Cartography and Cadastre Bureau).

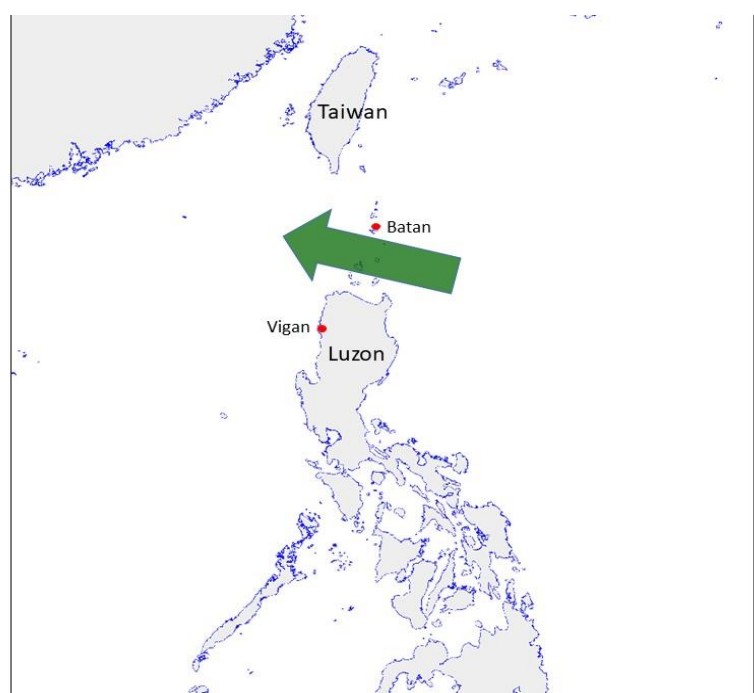

Figure 4. The possible track of Typhoon 1874 moving across the Luzon Strait between Batan and Vigan during the period from the evening on 21 September to the early morning on 22 September 1874.

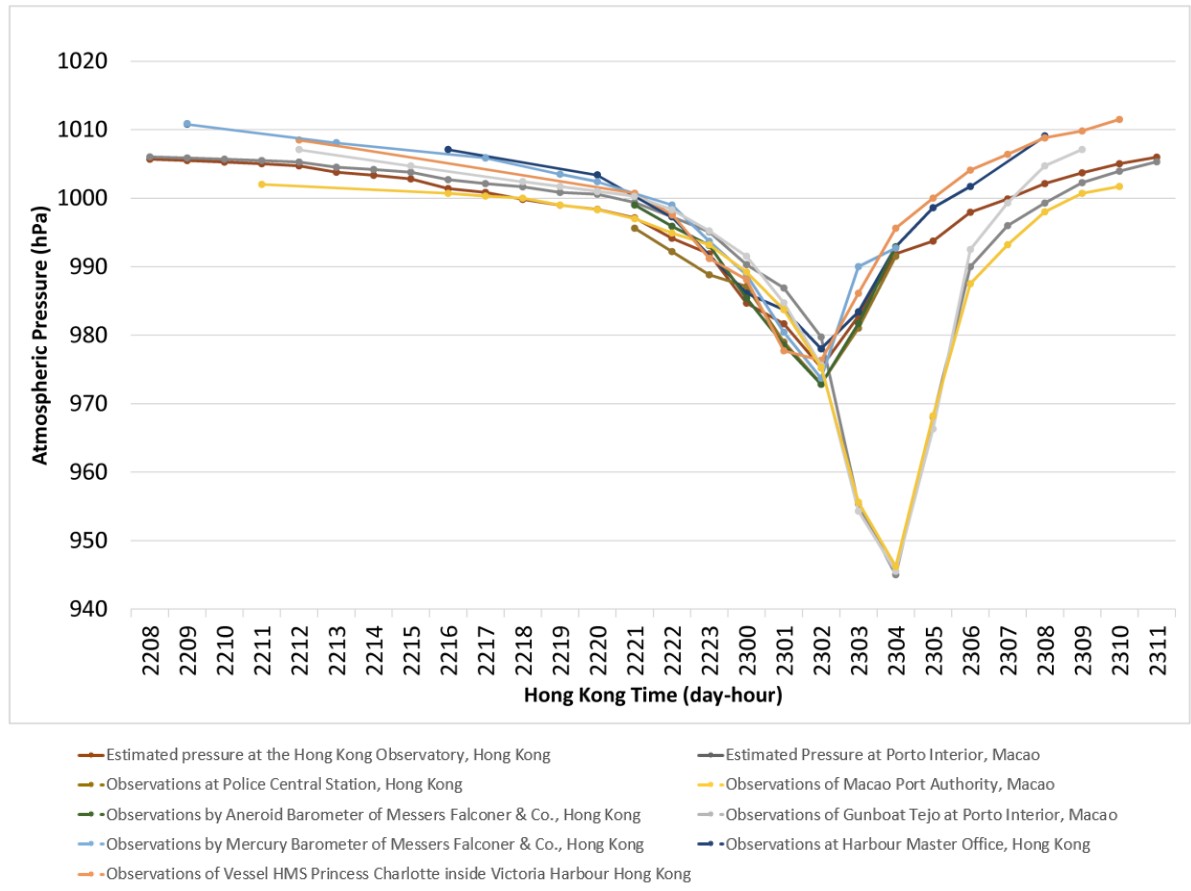

Figure 5. Series plots of pressure observations recorded at the Police Central Station, recorded by the Aneroid Barometer of Messers Falconer & Co., Hong Kong, the Mercury Barometer of Messers Falconer & Co., Hong Kong, Vessel HMS Princess Charlotte and Harbour Master Office in Hong Kong and recorded by the Macao Port Authority and Gunboat Tejo in Macao from 8 a.m. on 22 September to 11 a.m. on 23 September 1874 (as listed in Table 2) and the series plots of the atmospheric pressures at Hong Kong (Hong Kong Observatory) and Macao (Porto Interior) during the same period estimated by the Jelesnianski tropical cyclone model based on the hourly positions, hourly minimum mean sea level pressures near the centre and the radii of maximum wind of the reconstructed possible track as listed in Table 6.

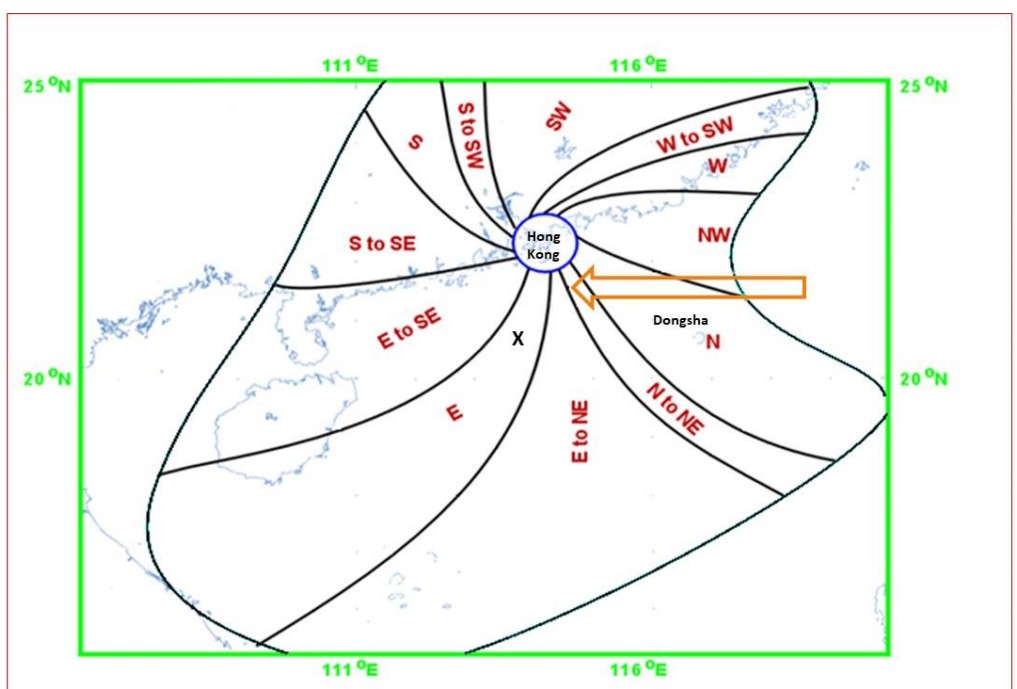

Figure 6. The 12-segment reference diagram showing the correlation between the wind direction at Waglan Island and tropical cyclone position during strong winds or above situations used by the Hong Kong Observatory (For example, the winds at Waglan Island will be easterly when the tropical cyclone is located at position '**X**').  The arrow framed in brown shows a typical track of tropical cyclone that could cause a sequential change of wind direction in Hong Kong similar to that of the passage of Typhoon 1874.

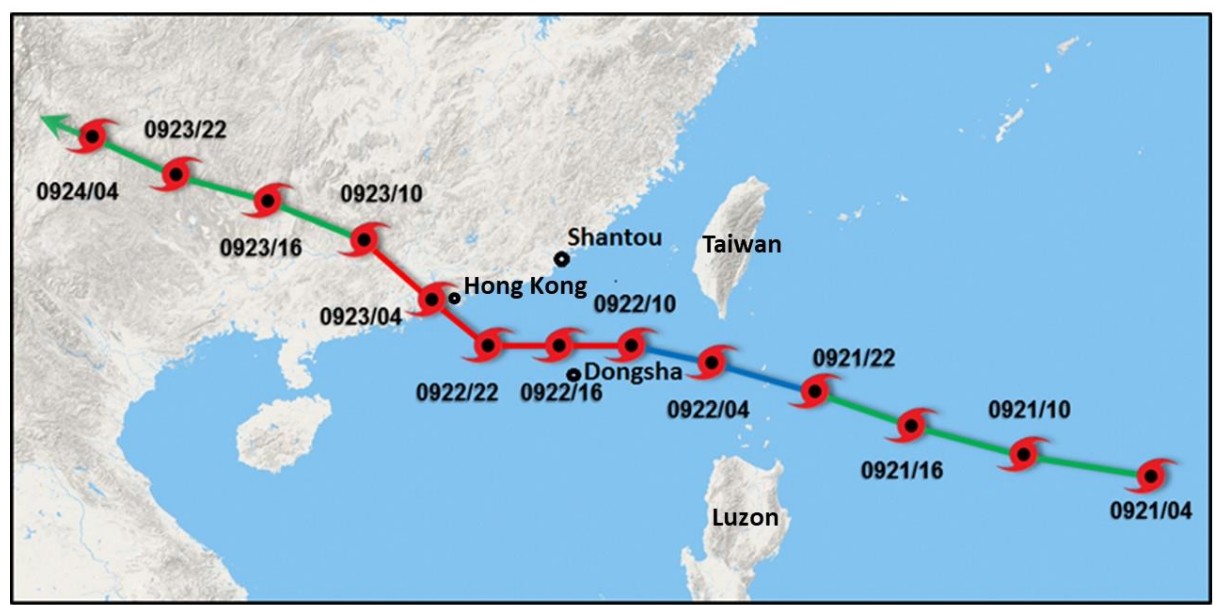

Figure 7 (a). The possible track of Typhoon 1874 passing through the Luzon Strait between Taiwan and Luzon and moving across the northern part of the South China Sea reconstructed in this study. Locations of Shantou and Dongsha are also shown.

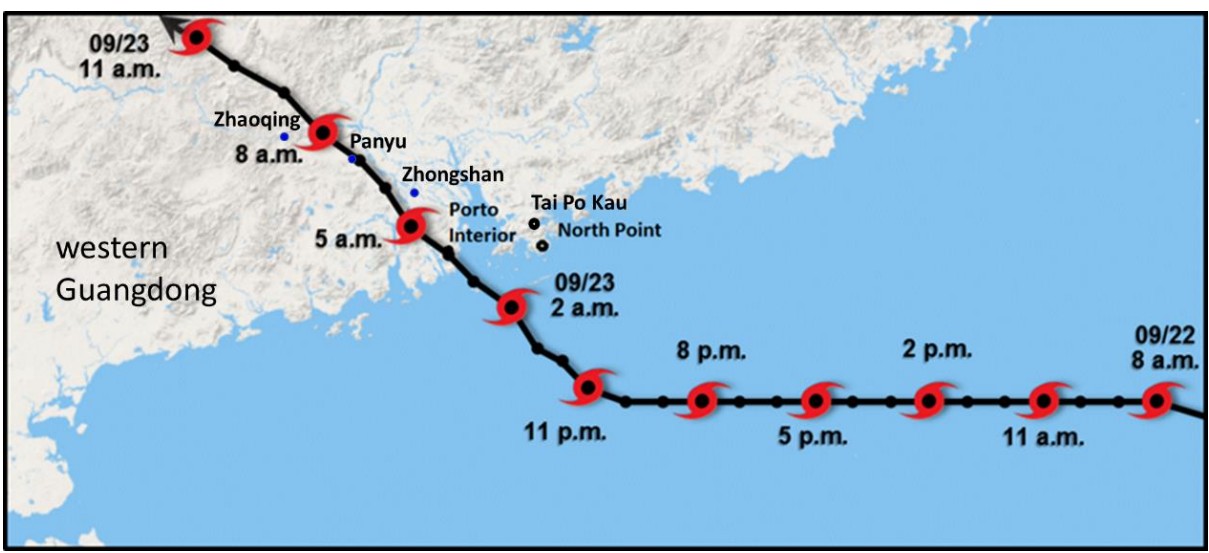

Figure 7 (b). The possible track of Typhoon 1874 moving along the coast of eastern Guangdong and the Pearl River Estuary reconstructed in this study. Locations of the tide gauges at North Point and Tai Po Kau in Hong Kong and Porto Interior in Macao, and locations of Zhongshan, Panyu and Zhaoqing in western Guangdong are also shown.

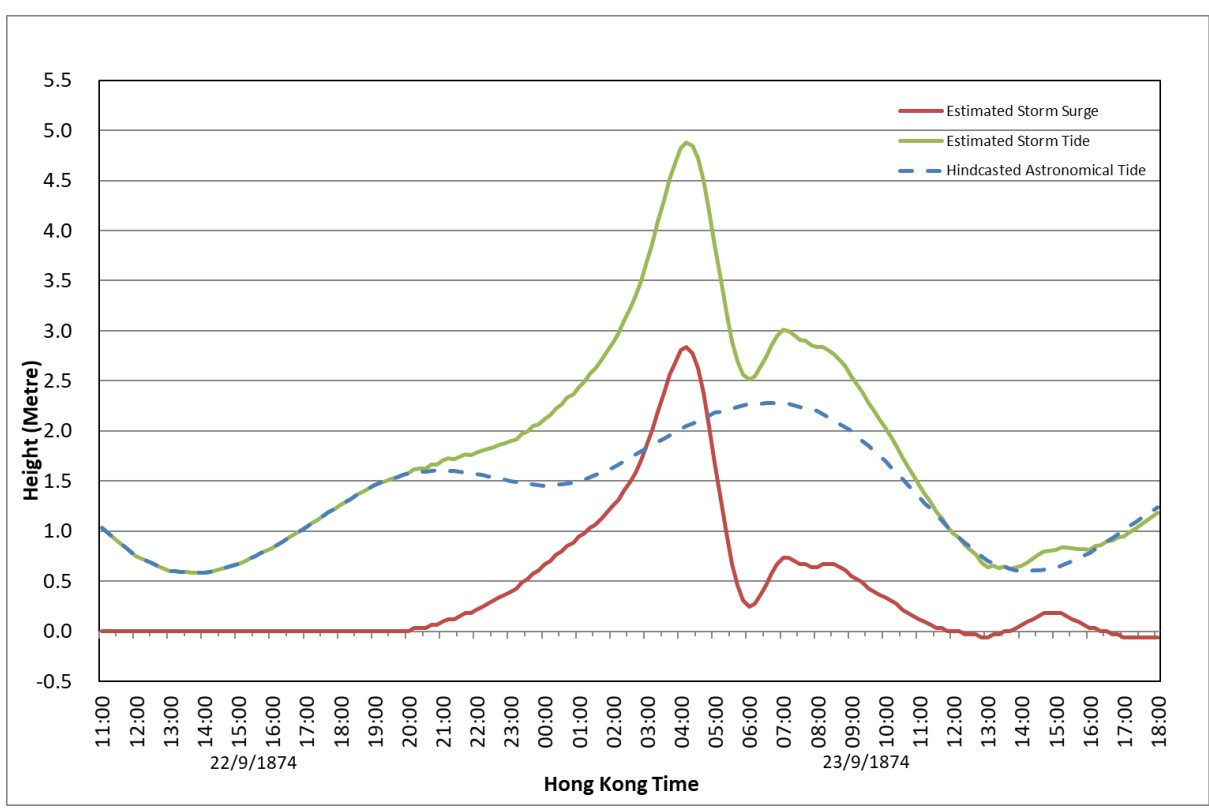

Figure 8. Time series of the hindcasted astronomical tide and estimated storm tide above Hong Kong Chart Datum and the estimated storm surge from 11 a.m. on 22 September to 6 p.m. on 23 September 1874 at North Point in Hong Kong. The estimated storm tide is the sum of the estimated storm surge and the astronomical tide.

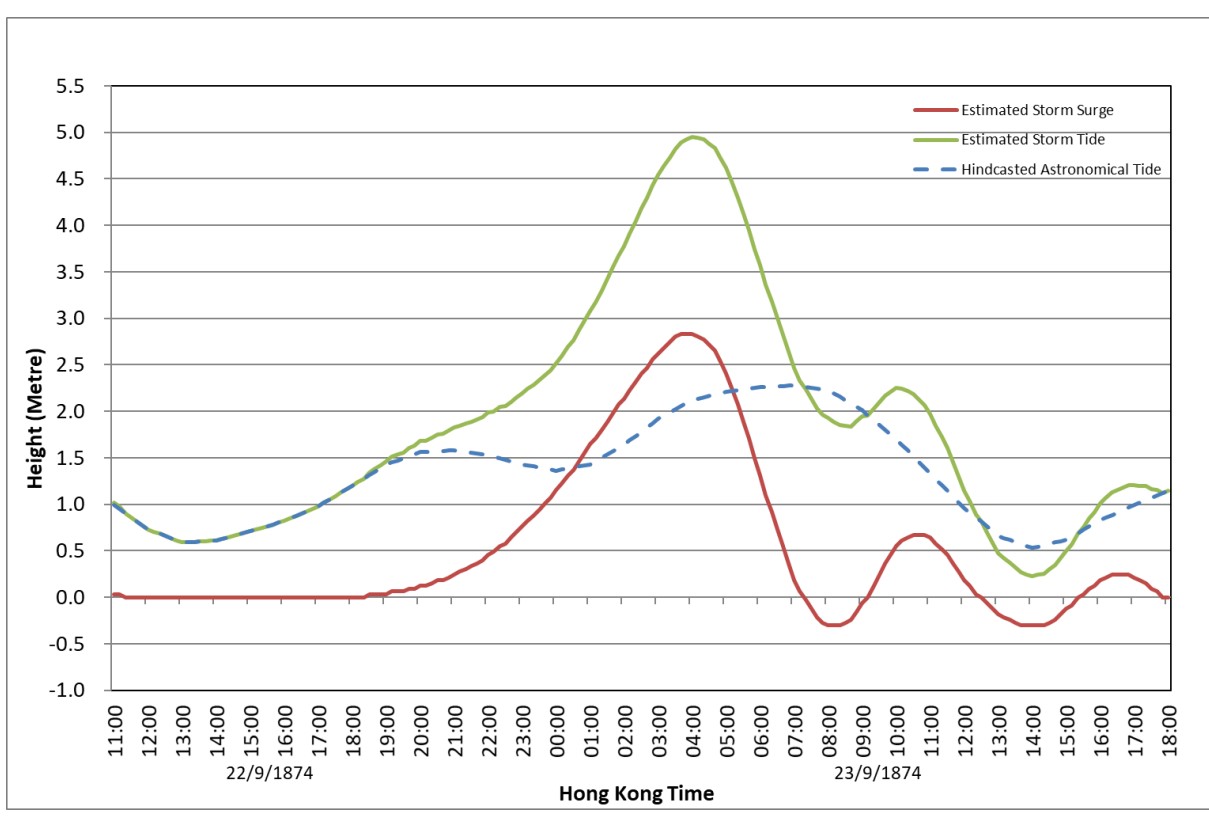

Figure 9. Time series of the hindcasted astronomical tide and estimated storm tide above Hong Kong Chart Datum and the estimated storm surge from 11 a.m. on 22 September to 6 p.m. on 23 September 1874 at Tai Po Kau in Hong Kong. The estimated storm tide is the sum of the estimated storm surge and the astronomical tide.

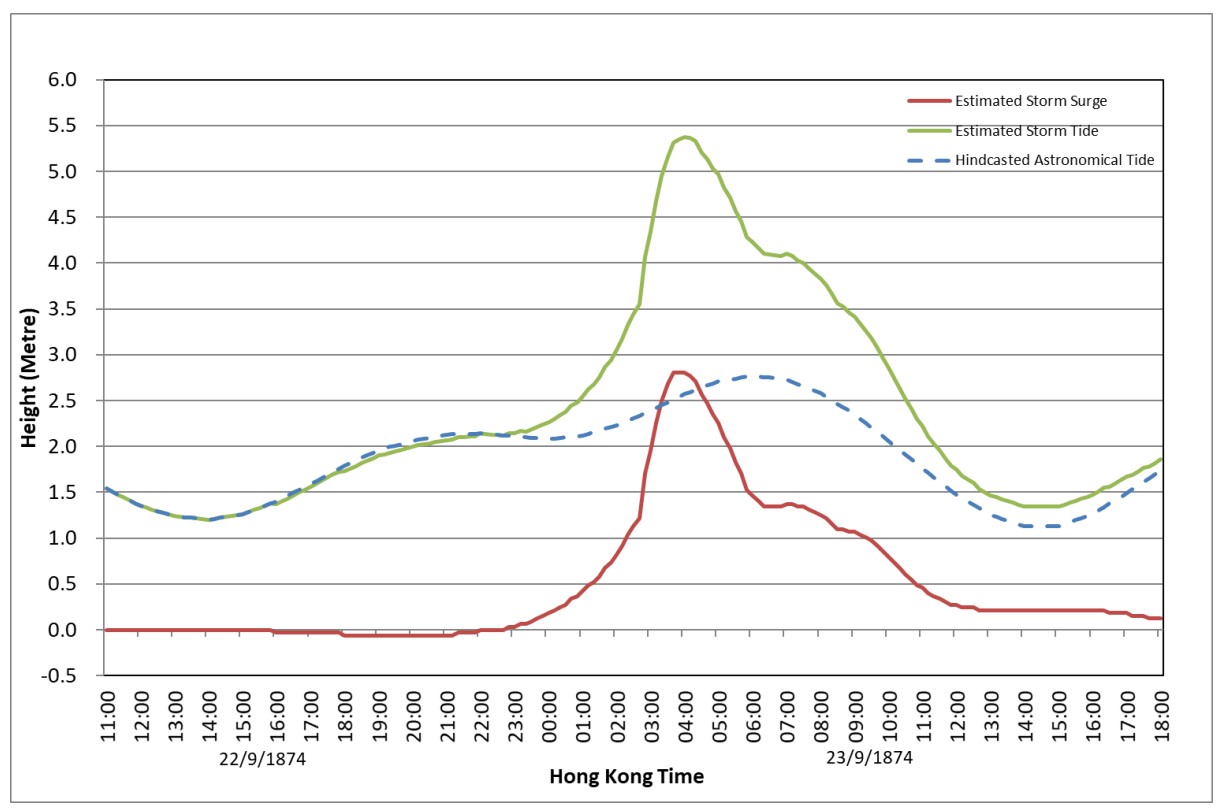

Figure 10. Time series of the hindcasted astronomical tide and estimated storm tide above Macao Chart Datum and the estimated storm surge from 11 a.m. on 22 September to 6 p.m. on 23 September 1874 at Porto Interior in Macao.  The estimated storm tide is the sum of the estimated storm surge and the astronomical tide.