# Peer review of "Reconstruction of track and simulation of storm surge associated with the calamitous typhoon affecting the Pearl River Estuary in September 1874"

_Climate of the Past, 2019_

## Referee Comment (RC1) · Anonymous Referee #1 · 5 Jun 2019

This is a very readable paper; it also presents a good case of data rescue and applies it to a potential hazard risk (storm surge/tide risk) estimate. The authors have made clear descriptions on the data collection, which is very good considering its historical nature. But I still have some concerns and suggestions for authors to rethink about their approach and propose for the study.

Based on the pressure and wind reports at four locations in HK and Macao, and information from Luzon, it is not difficult for one (with atmospheric training) to picture the likely track of the typhoon 1874 when approaching and passing this region. As authors also revealed, the track of typhoon 1874 was not uncommon. Of course, it is good and

encouraging that the authors are able to use further information and model to estimate and simulate the typhoon track and feed data into SLOSH to model maximum storm surge and tide of the typhoon. But what worried me the most are also the purpose for doing so and the great model uncertainty.

If my interpretation is correct, the authors want to use this super typhoon as said by them to exemplify that the potential storm surge and tide in HK and Macao can be much higher than what records have told us after 1883. If this is true, then the information is further important. However, given the topography and bathymetry they used are from the 1990s, I am afraid that the result could be overestimated and exaggerate the risk potential. Here are some reasons: (1) the numbers of height are way too higher than the descriptions in the historical documents and the time series pattern are also inconsistent. (2) The topography of Pearl River Estuary can change quickly in just several decades as other significant rivers in China, Taiwan and southeast Asia have revealed. Coastline changes usually accompany a large area of new tidal land which often experienced fast development and urbanization in the later centuries. Thus, if the purpose of the study is to give a realistic estimate of the storm surge/tide hazard risk, the authors need to seriously take this issue into account. Otherwise, it will be irresponsible to just give the tremendous numbers based on the simple model results. Also, I found it actually not difficult to find old HK maps dated back to the 19th century when doing a bit search myself.

In a word, this paper can potentially present a serious deficiency if the authors aim to simulate storm surge and tide brought by the typhoon 1874. This deficiency can result in overestimation and even distortion of the real situation. Further data especially those on topography and bathymetry are strongly recommended to be applied in the analysis. Or at least, the authors need to give a comprehensive explanation on the model results and give different scenarios (based on the further information) and uncertainty estimates.

Minor points: Figure 6 is unclear, further explanation is needed. Some detailed maps

are not necessary e.g. Figure 3. Figure 2 is provided without descriptions. Actually this can be used to explain present and 1800s topography in HK. Figure 8&9 are poorly explained: why the green and red curves drop down suddenly after peak and rise again after the down point by 6am (North Point)and then 8am (Tai Po Kau)? Line 78-80 on page 3 "which was on average 7 hours 36 minutes and 41 seconds ahead of UTC...". Can you explain why and how such exact time was calculated here?

––––––––––––––––––––––––––––––––

---

## Author Comment (AC1) · 10 Jun 2019

I would like to respond to the Referee Comments of Anonymous Referee #1 posted on 5 June 2019 as follows:

1. Given that change of topography and bathymetry will have effects to the storm surges, the effects could be overestimated or underestimated, depending on how the topography and bathymetry have changed. Hence, the use of the topography and bathymetry in the 1990's for running SLOSH could overestimate or underestimate the estimated storm surges for the typhoon in 1874. I agree that it would be highly desirable to run SLOSH using topography and bathymetry in the 1880's. However, while

locating maps of Hong Kong and Macao in the 1880's might not be that difficult, digitized bathymetry data with spatial resolution of about 1 km in Hong Kong and Macao waters and about 7 km in the open sea to the south of the Pearl River Estuary in the 1880's would very likely be not available for running SLOSH.

2. We have qualitatively discussed the possible sources (including the change in topography and bathymetry) of uncertainty of the estimated storm surges at Hong Kong and Macao for the typhoon in 'Results and Discussion' (Line 475 to Line 494) and stated that care has to be taken when comparing the storm surges and tides estimated in this study with those in other historical typhoons (Line 495 to Line 497). We have conducted a comparison of the SLOSH results using topography and bathymetry data in the 1990's and 2010's (not shown in the paper), during which quite a number of coastal development had occurred such as reclamations and building of new airports. The results show that the maximum storm surges at North Point, Tai Po Kau and Macao using topography and bathymetry data in the 1990's (2010's) are 2.83 m (2.71 m), 2.83 m (2.77 m) and 2.80 m (2.68 m) respectively. Perhaps this can give a brief quantitative idea on the sensitivity of SLOSH results on changes in topography and bathymetry.

3. Bearing the uncertainty of the estimated storm surges and tides, the reconstructed track (positions, intensities and radii of maximum winds) of the typhoon itself can be used as a possible scenario for assessment of storm surge risk in the Pearl River Estuary nowadays.

4. Comparison of the SLOSH results with descriptions in historical documents are described in the paragraph from Line 407 to Line 424. Regarding your comments on the inconsistency on 'the numbers of height are way too higher than the descriptions in the historical documents and the time series pattern are also inconsistent', I would like to elaborate further below:

(a) In Line 417, the historical document quotes 'By three, the water had risen to from five to six feet above its high water' meaning that the storm tide was 1.52 m (five feet)

to 1.83 m (six feet) above the astronomical high tide in Hong Kong. This is a bit higher than what is stated in Line 412 to Line 413 that 'the difference between the storm tide of 3.69 m at 3 a.m. and the astronomical high tide of 2.28 m at around 6:30 a.m.' - which is 1.41 m. Given the observation in the historical document is taken by human eyes at night time and not at the location of the tide station, such a small difference (1.52 m to 1.83 m against 1.41 m) is considered not inconsistent.

(b) In Line 422, the historical document quotes 'storm surge which caused severe flooding of up to 7 feet above high tide level' meaning that the maximum storm tide was up to 2.13 m (7 feet) above the high astronomical tide in Macao. This is smaller than what is stated in Line 418 to Line 419 that 'the difference between the maximum storm tide of 5.37 m at 4 a.m. and the astronomical high tide of 2.77 m at around 6 a.m.' – which is 2.60 m. Again, given the observation in the historical document is taken by human eyes at night time and not at the location of the tide station, such a difference (2.13 m against 2.60 m) is considered not inconsistent.

5. Response to minor comments:

(a) Figure 6 shows the prevailing wind directions in Hong Kong (for strong winds or above) with respect to tropical cyclone positions. The arrow shows a tropical cyclone track moving from east to west along the coast of south China to skirt the south of Hong Kong. The expected sequential change of the prevailing direction in Hong Kong will be NW, N, N to NE, E to NE, E, E to SE. This figure helps to support a westerly track of the typhoon in 1874 as discussed in Line 239 to Line 256.

(b) Figure 3 can be removed.

(c) The purpose of Figure 2 and 3 is to give the readers an idea on the locations in Hong Kong and Macao mentioned in the paper. These figures can be removed.

(d) The red line is the time series of storm surges simulated by SLOSH and the green line is the time series of storm tides (sum of storm surge and astronomical tide at the

Interactive
comment

same time). Sudden drop after peak and rise again after the down point is not uncommon for storm surges as storm surges at a particular location can change rapidly with changes in the distance of the tropical cyclone from the location, intensity of the tropical cyclone, storm size (in terms of radius of maximum winds) of the tropical cyclone and the prevailing wind direction. For typhoon 1874, the rapid drop after peak would be due to the fast departure (the typhoon was moving at a speed of about 38 km/hour (Line 327 to Line 329) and rapid weakening of the typhoon after making landfall (from 945 hPa at 4 a.m. to 980 hPa at 10 a.m. as shown in Table 5). The small rise again after the down point might be due to the change of the storm size (in terms of radius of maximum winds) from 25 km at 4 a.m. to 45 km at 10 a.m. as shown in Table 5 also. These discussions can be incorporated into the paper.

(e) The statement 'all times mentioned in thyis paper refer to the local mean time, which was on average 7 hours 36 minutes and 41 seconds ahead of UTC in Hong Kong before 1 November 1904' is a note to the readers that there was a change in the Hong Kong Time on 1 Nov 1904. Before 1 Nov 1904, Hong Kong Time was based on local solar time (i.e. sun's transit occurs at local solar noon), which according to the longitude of Hong Kong was on average Greenwich Mean Time (GMT) plus 7 hours 36 minutes and 41 seconds. After 1 Nov 1904, it was changed to GMT + 8 hours. It is just a note and will have no effect to the study results.

---

## Referee Comment (RC2) · Anonymous Referee #2 · 9 Jul 2019

[Evaluation] This manuscript presents very interesting research on reconstruction of storm surge which derives devastating casualties in the related regions. The methods used are appropriate and the conclusions derived from these and the interpretations are consistent and sound. I believe the paper will be of interest to the readership of this journal and would recommend it for acceptance after the minor revisions. I look forward to seeing it in print.

[Comment] I understand that the reconstructions of natural phenomena in historical time are very difficult because the information is very limited. However, I concern that reconstructed storm surge in this paper are derived from estimated typhoon truck.

[Figure]

So I would like to strongly recommend that the authors give more explanations about "reliability" of reduplication of reconstructions.

[Minor comment] Page 2, line 30-44: Please state criteria of typhoon, Super Typhoon and Severe Typhoon, because typhoon criteria may differ in several countries.

Figure 5: This figure will be better to be understood especially for readers outside of the region if topographical information in Macau is added.

Figure 5, 8, 9: In Figure captions, please provide detail explanations what colors of the lines mean.

---

## Author Comment (AC2) · 18 Jul 2019

Thanks for the comments of Anonymous Referee #2. We provide below a point-by-point response to the comments.

1. Anonymous Referee #2 [Comment]: I understand that the reconstructions of natural phenomena in historical time are very difiňĄcult because the information is very limited. However, I concern that reconstructed storm surge in this paper are derived from estimated typhoon truck. So I would like to strongly recommend that the authors give more explanations about "reliability" of reduplication of reconstructions.

[Figure]

Authors [Response]: Noted. The following paragraph will be added to 'Results and Discussion' after line 430 to elaborate more on how likely the reconstructed track is:

'It has to be noted that the parts of the reconstructed track over the western North Pacific and southwestern part of China (plotted in green in Figure 7(a)) were arbitrarily extended to meet the requirement of input of thirteen 6-hourly positions for running the storm surge model for estimation of storm surges in Hong Kong and Macao. The limited weather observations in the Luzon Strait area, though not sufficient enough to enable a detailed estimation of the positions of Typhoon 1874 moving over the Luzon Strait (plotted in blue in Figure 7(a)), had indicated that the typhoon had very likely moved across the Luzon Strait between Vigan and Batan with typhoon intensity on the early morning of 22 September. For the reconstructed track over the northeastern part of the South China Sea, the Pearl River Estuary and western Guangdong (plotted in red in Figure 7(a) and the whole track in Figure 7(b)), a quantitative and reliable estimation of the hourly positions, hourly minimum mean sea level pressures near the centre and the radii of maximum wind became possible by using the Jelesnianski tropical cyclone model based on the more comprehensive weather observations taken in Hong Kong and Macao. Besides reproducing well the trends of change of atmospheric pressure with time at Hong Kong and Macao, including the rapid fall of atmospheric pressure at Macao from 2 a.m. to 4 a.m. on 23 September while the atmospheric pressure at Hong Kong was rising (Figure 5), the atmospheric pressure readings taken at Hong Kong and Macao could also be reproduced quantitatively with reasonably small differ- ence. Comparing the hourly atmospheric pressures at Hong Kong and Macao during the period estimated by the Jelesnianski tropical cyclone model based on the hourly positions, hourly minimum mean sea level pressures near the centre and the hourly radii of maximum wind from 8 a.m. on 22 September to 11 a.m. on 23 September in Table 4 with the corresponding available hourly atmospheric pressure observations taken by the Hong Kong Harbour Master Office and Vessel HMS Princess Charlotte in Hong Kong (where the pressure readings were taken near mean sea level and closest to the Hong Kong Observatory) and Gunboat Tejo in Macao (where the pressure read-

ings were taken near mean sea level and at the Porto Interior), the root-mean-square of the differences were 4.0 hPa, 4.6 hPa and 2.7 hPa respectively. The differences were even smaller for the period from 8 p.m. on 22 September (when the storm surge at North Point started to rise and before the typhoon picked up a northwesterly track) to 4 a.m. on 23 September (when the storm surges at North Point, Tai Po Kau and Porto Interior were almost at the highest and the typhoon had made landfall) with root-mean-squares of 2.7 hPa, 3.1 hPa and 1.7 hPa respectively. Furthermore, the reconstructed track also matched well with the observed wind direction changes at Hong Kong during the approach and departure of the typhoon. Combining Figure 6 and Figure 7 could reveal that the wind direction at Hong Kong would veer gradually from northwesterly to northeasterly during the day on 22 September, and continue to veer to easterly and then southeasterly during the evening on 22 September and early morning of 23 September. This matched well with the observed wind direction changes reported by the Harbour Master and HMS Princess Charlotte as shown in Table 3. Such sequence of wind direction change would not occur if the typhoon approached Hong Kong from the southeast or south during the day on 22 September.'

2. Anonymous Referee #2 [Minor Comment]: Page 2, line 30-44: Please state criteria of typhoon, Super Typhoon and Severe Typhoon, because typhoon criteria may differ in several countries.

Authors [Response]: A link to the Hong Kong Observatory website showing classification of tropical cyclones (https://www.weather.gov.hk/informtc/class.htm) will be added as a footnote in Page 2.

3. Anonymous Referee #2: Figure 5: This figure will be better to be understood especially for readers outside of the region if topographical information in Macau is added.

Authors [Response]: I guess it is Figure 3, not Figure 5. We will try our best to replace Figure 3 with a map showing topographical information in Macao but copyright issue

would have to be resolved. If such a map with no copyright issue could not be found, we would like to keep Figure 3 as is as the main purpose of the map is to show the coastline (in particular, in the vicinity of Porto Interior) which is one of the major factors affecting storm surge.

4. Anonymous Referee #2: Figure 5, 8, 9: In Figure captions, please provide detail explanations what colors of the lines mean.

Authors [Response]: This will be done in the revised manuscript.

---

## Author Response (AR1)

[revised manuscript text omitted]

已刪除:

已刪除: and the chronological summary of the meteorological observations at Vigan and Batan Islands. Pressure observations had been converted from mmHg to hPa using 1 mm Hg = 1.33322387 hPa based on the Smithsonian Meteorological Tables, 4th revised edition, 1918

[Figure]

[Figure]

Estimated pressure at the Hong Kong Observatory, Hong Kong — Estimated Pressure at Porto Interior, Macao
Observations at Police Central Station, Hong Kong — Observations of Macao Port Authority, Macao
Observations by Aneroid Barometer of Messers Falconer & Co., Hong Kong — Observations of Gunboat Tejo at Porto Interior, Macao
Observations by Mercury Barometer of Messers Falconer & Co., Hong Kong — Observations at Harbour Master Office, Hong Kong
Observations of Vessel HMS Princess Charlotte inside Victoria Harbour Hong Kong

Figure 5. Series plots of pressure observations recorded at the Police Central Station, recorded by the Aneroid Barometer of Messers Falconer & Co., Hong Kong, the Mercury Barometer of Messers Falconer & Co., Hong Kong, Vessel HMS Princess Charlotte and Harbour Master Office in Hong Kong and recorded by the Macao Port Authority and Gunboat Tejo in Macao from 8 a.m. on 22 September to 11 a.m. on 23 September 1874 (as listed in Table 2) and the series plots of the atmospheric pressures at Hong Kong (Hong Kong Observatory) and Macao (Porto Interior) during the same period estimated by the Jelesnianski tropical cyclone model based on the hourly positions, hourly minimum mean sea level pressures near the centre and the radii of maximum wind of the reconstructed possible track as listed in Table 6.

[Figure]

[Figure]

已删除:

Figure 6. The 12-segment reference diagram showing the correlation between the wind direction at Waglan Island and tropical cyclone position during strong winds or above situations used by the Hong Kong Observatory (For example, the winds at Waglan Island will be easterly when the tropical cyclone is located at position 'X'). The arrow framed in brown shows a typical track of tropical cyclone that could cause a sequential change of wind direction in Hong Kong similar to that of the passage of Typhoon 1874.

[Figure]

[Figure]

已删除:

Figure 7 (a). The possible track of Typhoon 1874 passing through the Luzon Strait between Taiwan and Luzon and moving across the northern part of the South China Sea reconstructed in this study.  Locations of Shantou and Dongsha are also shown.

[Figure]

Figure 7 (b). The possible track of Typhoon 1874 moving along the coast of eastern Guangdong and the Pearl River Estuary reconstructed in this study. Locations of the tide gauges at North Point and Tai Po Kau in Hong Kong and Porto Interior in Macao, and locations of Zhongshan, Panyu and Zhaoqing in western Guangdong are also shown.

已删除: ,

[Figure]

Figure 8. Time series of the hindcasted astronomical tide and estimated storm tide above Hong Kong Chart Datum and the estimated storm surge from 11 a.m. on 22 September to 6 p.m. on 23 September 1874 at North Point in Hong Kong. The estimated storm tide is the sum of the estimated storm surge and the astronomical tide.

已刪除: .

已刪除: ,

已刪除: ( )

[Figure]

[Figure]

Figure 9. Time series of the hindcasted astronomical tide and estimated storm tide above Hong Kong Chart Datum and the estimated storm surge from 11 a.m. on 22 September to 6 p.m. on 23 September 1874 at Tai Po Kau in Hong Kong. The estimated storm tide is the sum of the estimated storm surge and the astronomical tide.

已删除: ,

已删除: ( )

[Figure]

[Figure]

[Figure]

已刪除:

Figure 10. Time series of the hindcasted astronomical tide and estimated storm tide above Macao Chart Datum and the estimated storm surge from 11 a.m. on 22 September to 6 p.m. on 23 September 1874 at Porto Interior in Macao. The estimated storm tide is the sum of the estimated storm surge and the astronomical tide.

已刪除: ,

已刪除: ( )

---

## Author Response (AR2)

Thanks for the comments of Anonymous Referee #1 and Anonymous Referee #3. We provide below a point-by-point response to the comments.

1. Anonymous Referee #1 [Minor suggestions]: This paper contains a large number of tables and figures. I suggest that some tables/figures can be merged or removed. For example, Table 2 and 3 can be merged into one table. And is Figure 4 necessary? The concept is simple or the authors can integrate the info into Figure 1 or 7.

   Authors [Response]: Noted. Table 2 and 3 were merged into one table. Figure 4 was removed and the locations of Vigan and Batan Islands were marked in Figure 6 in the revised manuscript. Furthermore, Table 2, Table 3 and Annex I were removed from the main text and put into the supplementary file.

2. Anonymous Referee #1 [Minor suggestions]: Also, line 220 "Dongsha as marked in Figure 1". However, Dongsha can not be found in the figure. Please check.

   Authors [Response]: "Dongsha" was added in Figure 1 in the revised manuscript.

3. Anonymous Referee #3 [Comments]: Table 1 and Figure 1: How are these typhoons selected? Do they represent all severe typhoons impacting HK since the establishment of HKO in 1883, or just those which happened to move across the Luzon Strait? The selection criteria should be defined.

   Authors [Response]: Thanks for pointing out the requirement of selection criteria. The manuscript was subsequently revised to set the selection criteria as top five storm surges and storm tides recorded up to 2018 since the establishment of the Hong Kong Observatory in 1883. In this respect, Super Typhoon Hope in 1979 and Severe Typhoon Hagupit in 2008 were removed from Table 1 and Figure 1 in the revised manuscript.

4. Anonymous Referee #3 [Comments]: RE: Section 2 and 3: Section 3 should probably be part of Section 2, as it describes the data sources of this study.

   Authors [Response]: Section 2 and 3 were merged into one section on "Data and Methods" in the revised manuscript.

5. Anonymous Referee #3 [Comments]: Line 141 and 210: Location of Dongsha is not shown in Figure 1.

Authors [Response]: "Dongsha" was added in Figure 1 in the revised manuscript.

6. Anonymous Referee #3 [Comments]: Annex I and Table 4 & 5 should be put into the supplementary file for the sake of typesetting.

   Authors [Response]: Table 2, Table 3 and Annex I were removed from the main text and put into the supplementary file in the revised manuscript.

7. Anonymous Referee #3 [Comments]: Figure 7a and 7b should be merged into one graph.

   Authors [Response]: Done in the revised manuscript.

8. Anonymous Referee #3 [Comments]: Figure 8-10 should be arranged into 3 panels in one graph.

   Authors [Response]: Done in the revised manuscript.

9. Anonymous Referee #3 [Comments]: There is no "Conclusion" section. Not sure if it is a must for CP papers.

   Authors [Response]: The primary objectives of this study are to reconstruct a possible track of Typhoon 1874 and estimate its associated storm surges and storm tides in Hong Kong and Macao during its passage using weather information/observations in historical documents. The outcomes of the study, including discussions on the possible uncertainties, are incorporated in the section on "Results and Discussion". Concluding remarks such as the importance and values of weather observations in historical documents and international joint effort on climatological data rescue and retrieval of the historical climate data to studies of historical weather events, and the importance of historical significant storm surge events in storm surge risk assessment are implicitly incorporated in the "Results and Discussion" section.

[revised manuscript text omitted]

Estimated pressure at the Hong Kong Observatory, Hong Kong — Estimated Pressure at Porto Interior, Macao
Observations at Police Central Station, Hong Kong — Observations of Macao Port Authority, Macao
Observations by Aneroid Barometer of Messers Falconer & Co., Hong Kong — Observations of Gunboat Tejo at Porto Interior, Macao
Observations by Mercury Barometer of Messers Falconer & Co., Hong Kong — Observations at Harbour Master Office, Hong Kong
Observations of Vessel HMS Princess Charlotte inside Victoria Harbour Hong Kong

Figure 4. Series plots of pressure observations recorded at the Police Central Station, recorded by the Aneroid Barometer of Messers Falconer & Co., Hong Kong, the Mercury Barometer of Messers Falconer & Co., Hong Kong, Vessel HMS Princess Charlotte and Harbour Master Office in Hong Kong and recorded by the Macao Port Authority and Gunboat Tejo in Macao from 8 a.m. on 22 September to 11 a.m. on 23 September 1874 (as listed in Table S1) and the series plots of the atmospheric pressures at Hong Kong (Hong Kong Observatory) and Macao (Porto Interior) during the same period estimated by the Jelesnianski tropical cyclone model based on the hourly positions, hourly minimum mean sea level pressures near the centre and the radii of maximum wind of the reconstructed possible track as listed in Table 3.

[Figure]

已删除:
Figure 4. The possible track of Typhoon 1874 moving across the Luzon Strait between Batan and Vigan during the period from the evening on 21 September to the early morning on 22 September 1874. .
.
.
.
.
.
.
已删除: <物件><物件>
已删除: 5
已删除: 2
已删除: 6

[Figure]

Figure 5. The 12-segment reference diagram showing the correlation between the wind direction at Waglan Island and tropical cyclone position during strong winds or above situations used by the Hong Kong Observatory (For example, the winds at Waglan Island will be easterly when the tropical cyclone is located at position 'X').  The arrow framed in brown shows a typical track of tropical cyclone that could cause a sequential change of wind direction in Hong Kong similar to that of the passage of Typhoon 1874.

已删除: 6

[Figure]

Figure 6. The possible track of Typhoon 1874 passing through the Luzon Strait between Taiwan and Luzon and moving across the northern part of the South China Sea reconstructed in this study.  Locations of Shantou, Dongsha, Vigan and Batan are also shown.  The small figure at the upper-right corner shows the part of the possible track of Typhoon 1874 before and after making landfall at the Pearl River Estuary and shows the locations of the tide gauges at North Point and Tai Po Kau in Hong Kong and Porto Interior in Macao, and locations of Zhongshan, Panyu and Zhaoqing in western Guangdong.

[Figure]

[Figure]

[Figure]

[Figure]

已删除:

Figure 7 (b). The possible track of Typhoon 1874 moving along the coast of eastern Guangdong and the Pearl River Estuary reconstructed in this study.  Locations of the tide gauges at North Point and Tai Po Kau in Hong Kong and Porto Interior in Macao, and locations of Zhongshan, Panyu and Zhaoqing in western Guangdong are also shown.  .
<物件>

已删除: 8

已删除: from 11 a.m. on 22 September to 6 p.m. on 23 September 1874

已删除: time series of

已删除: from 11 a.m. on 22 September to 6 p.m. on 23 September 1874

已删除:  .
.
<物件> .

Figure 9. Time series of the hindcasted astronomical tide and estimated storm tide above Hong Kong Chart Datum and the estimated storm surge from 11 a.m. on 22 September to 6 p.m. on 23 September 1874 at Tai Po Kau in Hong Kong.  The estimated storm tide is the sum of the estimated storm surge and the astronomical tide.  .
.
.
.
.
<物件> .

Figure 10. Time series of the hindcasted astronomical tide and estimated storm tide above Macao Chart Datum and the estimated storm surge from 11 a.m. on 22 September to 6 p.m. on 23 September 1874 at Porto Interior in Macao.  The estimated storm tide is the sum of the estimated storm surge and the astronomical tide.  .

Figure 7. Time series of the hindcasted astronomical tide and estimated storm tide above Hong Kong Chart Datum and the estimated storm surge at (a) North Point and (b) Tai Po Kau in Hong Kong, and the hindcasted astronomical tide and estimated storm tide above Macao Chart Datum and the estimated storm surge at (c) Porto Interior in Macao from 11 a.m. on 22 September to 6 p.m. on 23 September 1874.  The estimated storm tide is the sum of the estimated storm surge and the astronomical tide.

---

## Author Response (AR3)

Thanks for the comments of the Editor.    We provide below a point-by-point response to the comments.

1.  A reference to the SLOSH model is needed (and maybe also a link to where it can be obtained)

    Authors [Response]: Jelesnianski *et al.* (1992) is the reference to the SLOSH model and has already been quoted at Line 350 in the original manuscript.    A link to the SLOSH webpage operated by the National Oceanic and Atmospheric Administration (NOAA) of USA is added at Line 344 in the revised manuscript. However, as far as we understand, individual official request to NOAA is required to obtain SLOSH.

2.  I agree with one of the reviewers that a conclusion is lacking (see manuscript preparation guidelines: https://www.climate-of-the-past.net/for_authors/manuscript_preparation.html).    Please add a short paragraph after the results/discussion.

    Authors [Response]: 'Conclusion' is added after the results/discussion in the revised manuscript.

[revised manuscript text omitted]